# ARHGEF26 enhances *Salmonella* invasion and inflammation in cells and mice

**Jeffrey S. Bourgeois**[1,2]**, Liuyang Wang**[1]**, Agustin F. Rabino**[3]**, Jeffrey Everitt**[4]**, Monica I. Alvarez**[1]**, Sahezeel Awadia**[3]**, Erika S. Wittchen**[5]**, Rafael Garcia-Mata**[3]**, Dennis C. Ko**[1,2,6]*

**1** Department of Molecular Genetics and Microbiology, School of Medicine, Duke University, Durham, North Carolina, United States of America, **2** University Program in Genetics and Genomics, Duke University, Durham, North Carolina, United States of America, **3** Department of Biological Sciences, University of Toledo, Toledo, Ohio, United States of America, **4** Department of Pathology, Duke University Medical Center, Durham, North Carolina, United States of America, **5** Department of Cell Biology and Physiology, University of North Carolina, Chapel Hill, North Carolina, United States of America, **6** Division of Infectious Diseases, Department of Medicine, School of Medicine, Duke University, Durham, North Carolina, United States of America

* dennis.ko@duke.edu.

**Data Availability Statement:** All cellular GWAS data from H2P2 are available through the H2P2 web atlas hosted by Duke University Office of Information Technology (http://h2p2.oit.duke.edu/

## Abstract

*Salmonella* hijack host machinery in order to invade cells and establish infection. While considerable work has described the role of host proteins in invasion, much less is known regarding how natural variation in these invasion-associated host proteins affects *Salmonella* pathogenesis. Here we leveraged a candidate cellular GWAS screen to identify natural genetic variation in the *ARHGEF26* (*Rho Guanine Nucleotide Exchange Factor 26*) gene that renders lymphoblastoid cells susceptible to *Salmonella* Typhi and Typhimurium invasion. Experimental follow-up redefined ARHGEF26's role in *Salmonella* epithelial cell infection. Specifically, we identified complex serovar-by-host interactions whereby ARHGEF26 stimulation of *S*. Typhi and *S*. Typhimurium invasion into host cells varied in magnitude and effector-dependence based on host cell type. While ARHGEF26 regulated SopB- and SopE-mediated *S*. Typhi (but not *S*. Typhimurium) infection of HeLa cells, the largest effect of ARHGEF26 was observed with *S*. Typhimurium in polarized MDCK cells through a SopB- and SopE2-independent mechanism. In both cell types, knockdown of the ARHGEF26-associated protein DLG1 resulted in a similar phenotype and serovar specificity. Importantly, we show that ARHGEF26 plays a critical role in *S*. Typhimurium pathogenesis by contributing to bacterial burden in the enteric fever murine model, as well as inflammation in the colitis infection model. In the enteric fever model, SopB and SopE2 are required for the effects of *Arhgef26* deletion on bacterial burden, and the impact of *sopB* and *sopE2* deletion in turn required ARHGEF26. In contrast, SopB and SopE2 were not required for the impacts of *Arhgef26* deletion on colitis. A role for ARHGEF26 on inflammation was also seen in cells, as knockdown reduced IL-8 production in HeLa cells. Together, these data reveal pleiotropic roles for ARHGEF26 during infection and highlight that many of the interactions that occur during infection that are thought to be well understood likely have underappreciated complexity.

H2P2Home/). All other relevant data are within the manuscript and its Supporting Information files.

**Funding:** This research was supported by National Institutes of Health 1F31AI143147 awarded to JSB, National Institutes of Health R01AI118903 and R21AI144586 awarded to DCK, and National Institutes of Health R01GM136826 awarded to RGM. The funders played no role in the study design, data collection and analysis, decision to publish, or preparation of the manuscript.

**Competing interests:** The authors have declared that no competing interests exist.

## Author summary

During infection, *Salmonella* manipulates host cells into engulfing the bacteria and establishing an intracellular niche. While many studies have identified genes involved in different stages of this *Salmonella* invasion process, few studies have examined how differences between human hosts contribute to infection susceptibility. Here we leveraged a candidate genetic screen to identify natural genetic variation in the human *ARHGEF26* gene that correlates with *Salmonella* invasion. Springboarding from this result, we experimentally tested and redefined ARHGEF26's role in *Salmonella* invasion, discovered a new role for ARHGEF26 in regulating inflammation during *Salmonella* disease, and demonstrated the relevance of these findings in mouse models. Building on how ARHGEF26 functions in other contexts, we implicated two ARHGEF26-interacting host proteins as contributors to *Salmonella* pathobiology. Collectively, these results identify a potential source of inter-person diversity in susceptibility to *Salmonella* disease and expand our molecular understanding of *Salmonella* infection to include a multifaceted role for ARHGEF26. They further identify important future directions in understanding how *Salmonella* recruit and manipulate ARHGEF26 as well as how ARHGEF26 is able to drive *Salmonella*-beneficial processes.

## Introduction

The ability for bacteria to invade non-phagocytic host cells has long been recognized as a crucial trait of many pathogenic bacteria. Observations of this phenomena in *Salmonella* stretch back to at least 1920 when Margaret Reed Lewis observed that *Salmonella enterica* serovar Typhi (*S.* Typhi) induces vacuole formation during invasion of chick embryo tissues [1]. With the advent of molecular biology, Galán and others identified that the type-III secretion system coded by genes in the *Salmonella* Pathogenicity Island-1 (SPI-1) facilitates *Salmonella* invasion [2,3]. Additional work demonstrated that the *Salmonella* effector proteins SopB, SopE, and SopE2 drive uptake of *Salmonella* in cultured cells by macropinocytosis through their ability to hijack or mimic host proteins to cause membrane ruffling [4–9]. The importance of *Salmonella* invasion has been affirmed through several *in vivo* studies, as strains defective for the invasion apparatus are severely attenuated in their ability to colonize and disseminate [3] and/or drive inflammation [10,11] in mouse models.

While much is known about the molecular mechanisms of host-*Salmonella* interactions, significantly less is known about why individuals have different susceptibilities to *Salmonella* infection. For example, in one recent *S.* Typhi human challenge study, the amount of *S.* Typhi found in patient blood varied substantially (0.05–22.7 CFU/mL blood in unvaccinated patients), and 23% of participants naturally resisted Typhoid fever onset [12]. To help fill this gap, genome-wide association studies (GWAS) of Typhoid fever and non-typhoidal *Salmonella* bacteremia have demonstrated the importance of the HLA-region [13] and immune signaling [14] in *Salmonella* susceptibility. We hypothesized that differences in susceptibility to SPI-1 effectors and *Salmonella* host cell invasion also regulate risk of *Salmonella* infection. In fact, using a novel cellular genome-wide association platform called Hi-HOST [15–17], we previously determined that SNPs that affect *VAC14* expression regulate susceptibility to *S.* Typhi invasion through regulation of plasma membrane cholesterol [18]. This demonstrates the power of cellular GWAS to identify natural genetic variation in cellular traits and enhance our mechanistic understanding of variable disease susceptibility.

In this work, we leveraged current understanding of host factors manipulated by *Salmonella* to identify human genetic variation that regulates *Salmonella* uptake into host cells. We identified a locus in the guanine exchange factor (GEF) *ARHGEF26* (also known as *SGEF*) that correlated with susceptibility to *S*. Typhi and *S*. Typhimurium invasion. Previous work has demonstrated that ARHGEF26 contributes to *Salmonella*-induced membrane ruffling and was hypothesized to impact invasion [7]. Current models speculate that ARHGEF26 contributes to membrane ruffling by enabling SopB-mediated activation of the human small GTPase RHOG. Here we demonstrated that ARHGEF26 regulates susceptibility to *Salmonella* uptake through *ARHGEF26* knockdown and overexpression. We also expanded our understanding of *Salmonella*'s interaction with ARHGEF26, finding that *S*. Typhi, but not *S*. Typhimurium uses ARHGEF26, DLG1 (also known as SAP97) and SCRIB (also known as Scribble) for SopB- and SopE- mediated infection of HeLa cells. Notably, we observed no effect of *RHOG* knockdown on infection, in line with recent studies that found RHOG is dispensable for invasion [19,20]. In contrast, we found that in polarized MDCK epithelial cells, ARHGEF26 and DLG1 contribute to *S*. Typhimurium but not *S*. Typhi invasion, independent of both SopB and SopE2. Following up on these *in vitro* studies, we found that *Arhgef26* deletion restricted *S*. Typhimurium burden in an enteric fever model of infection. This result was not only dependent on the effectors SopB and SopE2, but we note that SopB and SopE2 require ARHGEF26 to promote *S*. Typhimurium fitness in the murine gut. Finally, we report a previously unappreciated role for ARHGEF26 in regulating the inflammatory response to *Salmonella* in both HeLa cells and mice. Collectively, these data identify a novel locus that contributes to natural genetic susceptibility to SPI-1-mediated invasion and elucidate our mechanistic understanding of ARHGEF26-mediated *Salmonella* uptake and inflammation.

# Results

## Cellular GWAS identifies an association between the rs993387 locus and host cell susceptibility to *Salmonella* invasion

Our previous cellular GWAS (hereafter called H2P2; [17]) linked natural human genetic variation across 528 genotyped lymphoblastoid cell lines (LCLs) from parent-offspring trios to 79 infection phenotypes, including rates of *Salmonella* invasion. To quantify *Salmonella* invasion in H2P2, we infected LCLs with GFP-tagged *Salmonella enterica* serovar Typhi (*S*. Typhi) or *Salmonella enterica* serovar Typhimurium (*S*. Typhimurium) for one hour before gentamicin treatment. After an additional hour, we added IPTG to induce bacterial GFP expression, and three hours post infection we counted the GFP+ host cells, which contain viable bacteria, by flow cytometry (Fig 1A). Using the percent of host cells harboring viable *Salmonella* as a quantitative trait, we performed GWAS to identify loci associated with susceptibility to *Salmonella* invasion. Importantly, this assay differs from traditional measures of invasion as it focuses on bacterial uptake as a binary from the host perspective and, as a result, is insensitive to cooperative invasion or early replication events.

H2P2 identified 17 SNPs that passed genome-wide significance ($p < 5 \times 10^{-8}$), however, no *Salmonella* invasion-associated SNPs passed this threshold [17]. We next leveraged the last twenty years of *Salmonella* cellular microbiology and restricted our search space to common SNPs (minor allele frequency > 0.05) in 25 genes that regulate *Salmonella*-induced actin rearrangement, membrane ruffling, and/or invasion (Fig 1B, Table A in S1 Tables). These host genes encode proteins affected by SPI-1 secreted proteins (reviewed [21,22]), and include *ARF1* [23], *ARF6* [23,24], *ARHGEF26* (commonly called *SGEF*) [7], *RHOG* [7,25], *CYTH2* [23,24], *CDC42* [7,26–28], *RAC1* [7,26,28,29], and actin (*ACTB*) [30–33], as well as genes in

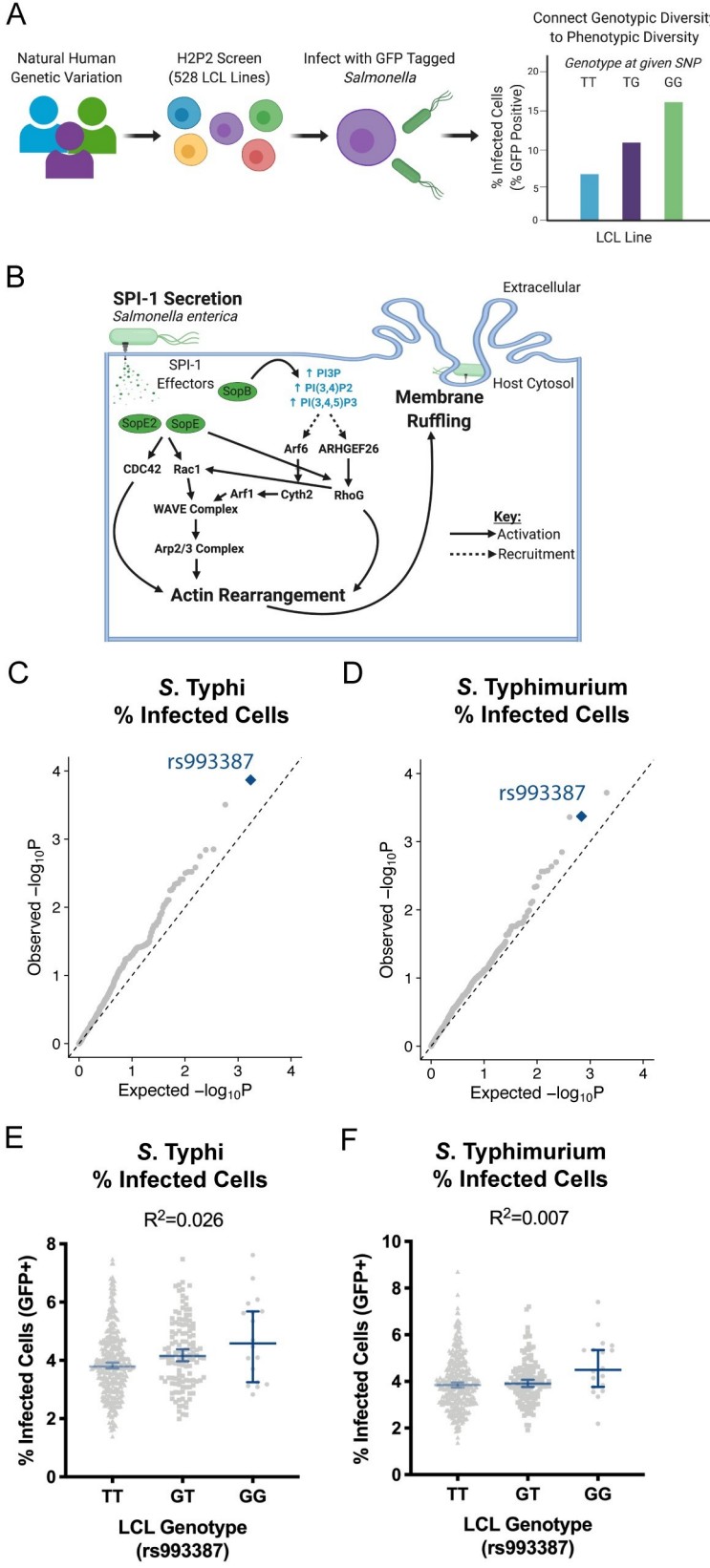

**Fig 1. H2P2 reveals the rs993387 locus is associated with lymphoblastoid cell line susceptibility to *Salmonella* invasion.** (A) Schematic for the H2P2 cellular GWAS. 528 lymphoblastoid cell lines (LCLs) from four populations were infected with *S*. Typhimurium (MOI 30) or *S*. Typhi (MOI 5) for 1 hour. Uptake was quantified 3 hours post infection by flow cytometry. Percent Infected Cells (GFP+) was used as a phenotype for GWAS analysis. (B) Schematic for SPI-1-mediated invasion. Genes and complexes listed are included in the stratified GWAS analysis. (C,D) Stratified QQ plots examining SNPs associated with *S*. Typhi (C) and *S*. Typhimurium (D) uptake. Only SNPs in SPI-1 invasion-associated host genes were considered and analysis was restricted to common SNPs (MAF>0.05) and pruned at $r^2$>0.6. rs993387 (blue diamond) diverges from p-values expected by chance for both serovars. Empirical P-values were calculated from family-based association analysis using QFAM-parents in PLINK. (E,F) Analysis of percent infected cells for *S*. Typhi (E) and *S*. Typhimurium (F) from the H2P2 screen plotted by rs993387 genotype. Each dot represents a single LCL line, averaged between three independent experiments. Bar marks the median and the error bars represent the 95% confidence intervals. $R^2$ values derived from simple linear regression. P values for both regressions were p≤0.05.

the WAVE [23,28,34] and Arp2/3 [27,28,34] complexes. Together, the proteins in this cascade lead to actin cytoskeletal rearrangements that enable macropinocytosis and bacterial uptake.

Plotting this SNP subset on a QQ plot to compare expected and observed p-values revealed a deviation towards p-values lower than expected by chance for both *S*. Typhi (Fig 1C) and *S*. Typhimurium (Fig 1D) invasion. The lowest p-value SNP associated with *S*. Typhi invasion was rs993387 (p = 0.0001), which is located in an *ARHGEF26* intron. This SNP also showed deviation with susceptibility to *S*. Typhimurium invasion (p = 0.0004), although a linked SNP, rs71744878 (LD $r^2$ = 0.63 for ESN; 0.48 for GWD; 0.97 for KHV; 0.75 for IBS from LD Link [35]) had a slightly lower p-value. Removing all *ARHGEF26* SNPs from the analysis returned the remaining SNPs to the expected neutral distribution, suggesting, surprisingly, we only detect natural genetic variation in *ARHGEF26* that substantially impacted *Salmonella* uptake (S1A and S1B Fig). The rs993387 G allele associated with susceptibility to invasion, and the SNP appears to have a larger effect on *S*. Typhi invasion (2.6%, Fig 1E) compared to *S*. Typhimurium invasion (0.7%, Fig 1F). Within each of the four populations used in H2P2, the rarity of the minor allele made it difficult to comment on the impact of the GG genotype, but the directionality of effect between the TT and GT genotypes was preserved across all populations (S1C Fig).

## Manipulating *ARHGEF26* expression phenocopies the rs993387 locus' effect on *Salmonella* uptake

We next analyzed the rs993387 locus in detail and found that SNPs in a ~100kb region of linkage disequilibrium overlapping *ARHGEF26* were associated with both *S*. Typhi (Fig 2A) and *S*. Typhimurium (Fig 2B) invasion. Looking for plausible functional variants in high LD ($r^2$ > 0.6 in EUR and AFR populations) in Haploreg [36] revealed only additional intronic variants. Further evaluation of published eQTL datasets [37,38] did not reveal a definitive connection to *ARHGEF26* mRNA expression. Testing for enhancer activity of a ~5kb region including rs993387, exon 11, and rs2122363 (the SNP with the lowest p-value for *S*. Typhimurium invasion (Fig 2B) using a luciferase reporter plasmid [39] demonstrated roughly two-fold induction over the vector control but no allele-specific enhancer activity in HeLa cells (S1D Fig). Additional analysis of the GTEx database [37] revealed that rs993387 is a splicing QTL in multiple tissues, including the colon (p = $1.1 \times 10^{-9}$), representing a plausible mechanism by which this SNP could regulate *ARHGEF26* protein abundance or function. In summary, while H2P2 implicated the *ARHGEF26* region in regulating susceptibility to *Salmonella* invasion, we do not yet know how genetic variation in this region affects *ARHGEF26* expression and/or function.

We next examined if *ARHGEF26* expression affects *Salmonella* invasion into LCLs. While previous reports have linked *ARHGEF26* to the induction of membrane ruffling [7], no study

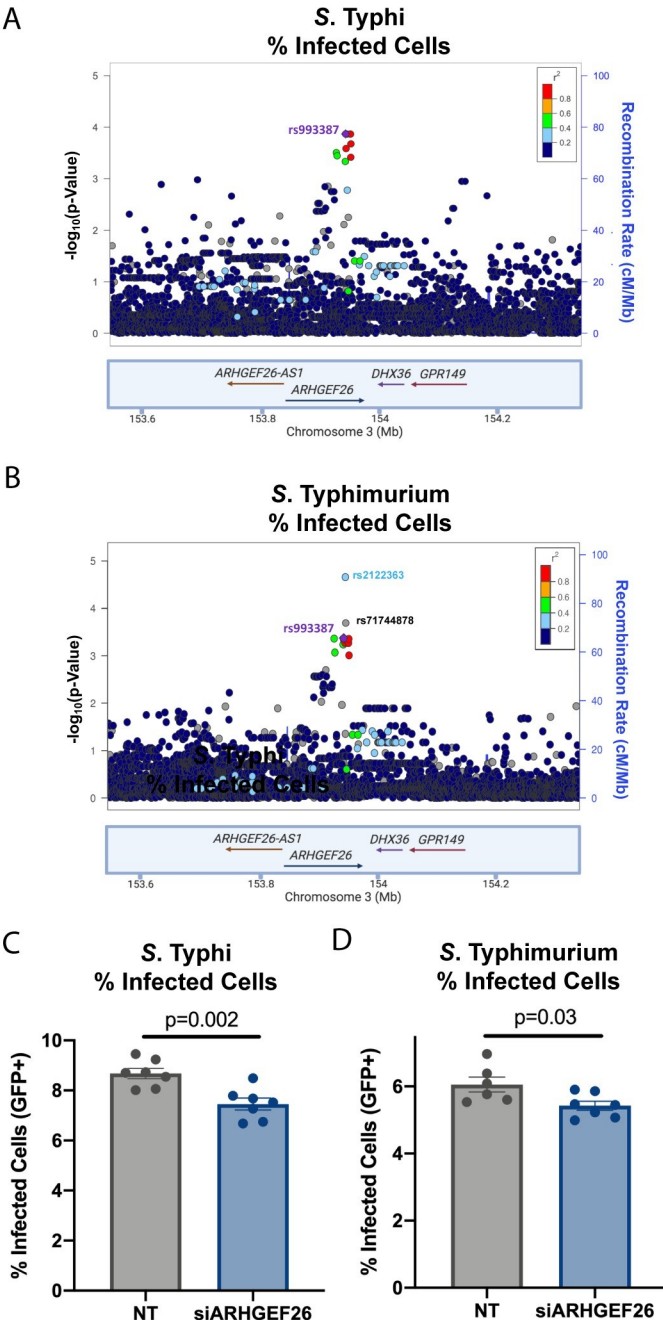

**Fig 2. Knockdown of ARHGEF26 phenocopies human genetic variation in the rs993387 locus.** (A,B) LocusZoom [41] plot generated with H2P2 data show SNPs in linkage disequilibrium with rs993387 in the *ARHGEF26* gene associated with the percent of host cells infected with *S*. Typhi (A) and *S*. Typhimurium (B). Height of dots represent the -$\log_{10}$(p-value) from H2P2. Dot color represents linkage disequilibrium ($r^2$) based on 1000 Genomes African dataset. Blue line behind dots tracks the recombination rate. (C,D) RNAi-mediated *ARHGEF26* knockdown reduces *S*. Typhi uptake into LCLs compared to non-targeting (NT) siRNA (siARHGEF26/NT = 0.86, C) and *S*. Typhimurium uptake (siARHGEF26/NT = 0.90, D). LCL line used was HG01697, from the IBS population, with the rs993387 genotype = GG). Cells were infected at MOI 5 (*S*. Typhi) or MOI 30 (*S*. Typhimurium) for 60 minutes. Percent infected cells was measured three hours post infection by flow cytometry. Each dot represents a biological replicate from three independent experiments. The reduction of *ARHGEF26* expression in these experiments was ≥40% based on qPCR. Experimental means were normalized to the grand mean prior to plotting or performing statistics. Bars represent the mean and error bars represent standard error of the mean. For C and D, p-values generated by an unpaired t-test.

has demonstrated whether *ARHGEF26* contributes to uptake into the host cell. This is an important distinction, as invasion does not always correlate with ruffling [40]. RNAi knockdown of *ARHGEF26*, confirmed by qPCR (knockdown ≥40%, S1E Fig), showed reduced proportion of LCLs with *S.* Typhi and *S.* Typhimurium uptake following infection—a phenotype similar to the protective rs993387 T-allele in LCLs (Fig 2C and 2D).

## ARHGEF26 increases susceptibility of HeLa cells to *S.* Typhi, but not *S.* Typhimurium, invasion

To dissect how ARHGEF26 contributes to *Salmonella* invasion, we examined how the protein regulates *Salmonella* invasion of HeLa cells, a common epithelial invasion model. We hypothesized that our core phenotypes of *ARHGEF26* positively regulating *S.* Typhi and *S.* Typhimurium invasion in LCLs would replicate in HeLa cells. *ARHGEF26* RNAi knockdown (~75% reduction, S1F Fig) significantly reduced the proportion of cells susceptible to *S.* Typhi invasion (Fig 3A). Notably this phenotype was larger than we observed with LCL knockdown. While a number of factors could contribute to this difference, we observed that LCLs expressed *ARHGEF26* close to the technical limit of our TaqMan assay, while HeLa cells expressed easily detectable levels of *ARHGEF26*. We therefore hypothesize that increased ARHGEF26 expression in HeLa cells may promote a larger role for the protein in invasion.

The prevailing model for *ARHGEF26* involvement in invasion is that Salmonellae use SopB to recruit ARHGEF26 and thereby activate RHOG, while SopE and SopE2 independently and directly activate RHOG [7]. Under this model, ARHGEF26 should only affect SopB-mediated invasion. To test this hypothesis, we infected HeLa cells with either a Δ*sopB* (membrane ruffling occurs through SopE) or Δ*sopE* (membrane ruffling occurs through SopB) strain of *S.* Typhi. Surprisingly, *ARHGEF26* knockdown reduced the percentage of infected cells when either effector was deleted, indicating ARHGEF26 regulation of both SopB- and SopE- mediated invasion (Fig 3B). Only when both SopB and SopE were deleted did *ARHGEF26* knockdown have no effect on *S.* Typhi invasion, though a higher MOI was required to observe reliable invasion (Fig 3C). Together, these data suggest that the previous model in which ARHGEF26 is recruited by only SopB (and functionally replaced by SopE) is incomplete, and instead support a model in which ARHGEF26 contributes to both SopB- and SopE-mediated invasion.

In striking contrast to our results in LCLs, while *ARHGEF26* knockdown protected HeLa cells from *S.* Typhi invasion, it did not impact the percentage of cells that were infected with *S.* Typhimurium (Fig 3D). This suggests that there is a complex host by serovar interaction that impacts whether ARHGEF26 is manipulated by *Salmonella* to promote invasion. Based on this, we speculated that SopE2 (present in *S.* Typhimurium 14028s; absent in *S.* Typhi Ty2) might be a more potent activator of host GTPases in HeLa cells than SopE (present in *S.* Typhi Ty2; absent in *S.* Typhimurium 14028s), and that this effector repertoire difference might explain our difference in *ARHGEF26*-dependent invasion. Supporting this theory, complementing our Δ*sopB*Δ*sopE S.* Typhi with an *S.* Typhimurium *sopB* gene restored the impact of *ARHGEF26* knockdown on uptake susceptibility, while expressing *S.* Typhimurium *sopE2* actually increased invasion of host cells with *ARHGEF26* knockdown to levels higher than those transfected with non-targeting siRNA (Fig 3E). However, in an *S.* Typhimurium strain lacking SopE2 (leaving only SopB to drive invasion), there was no effect of *ARHGEF26* knockdown on *S.* Typhimurium uptake (Fig 3F). This suggests that while SopE2 appears sufficient to overcome *ARHGEF26* knockdown, additional differences in effector repertoires and/or invasion mechanisms between Ty2 *S.* Typhi and 14028s *S.* Typhimurium are necessary to explain the differential reliance on *ARHGEF26* to invade HeLa cells.

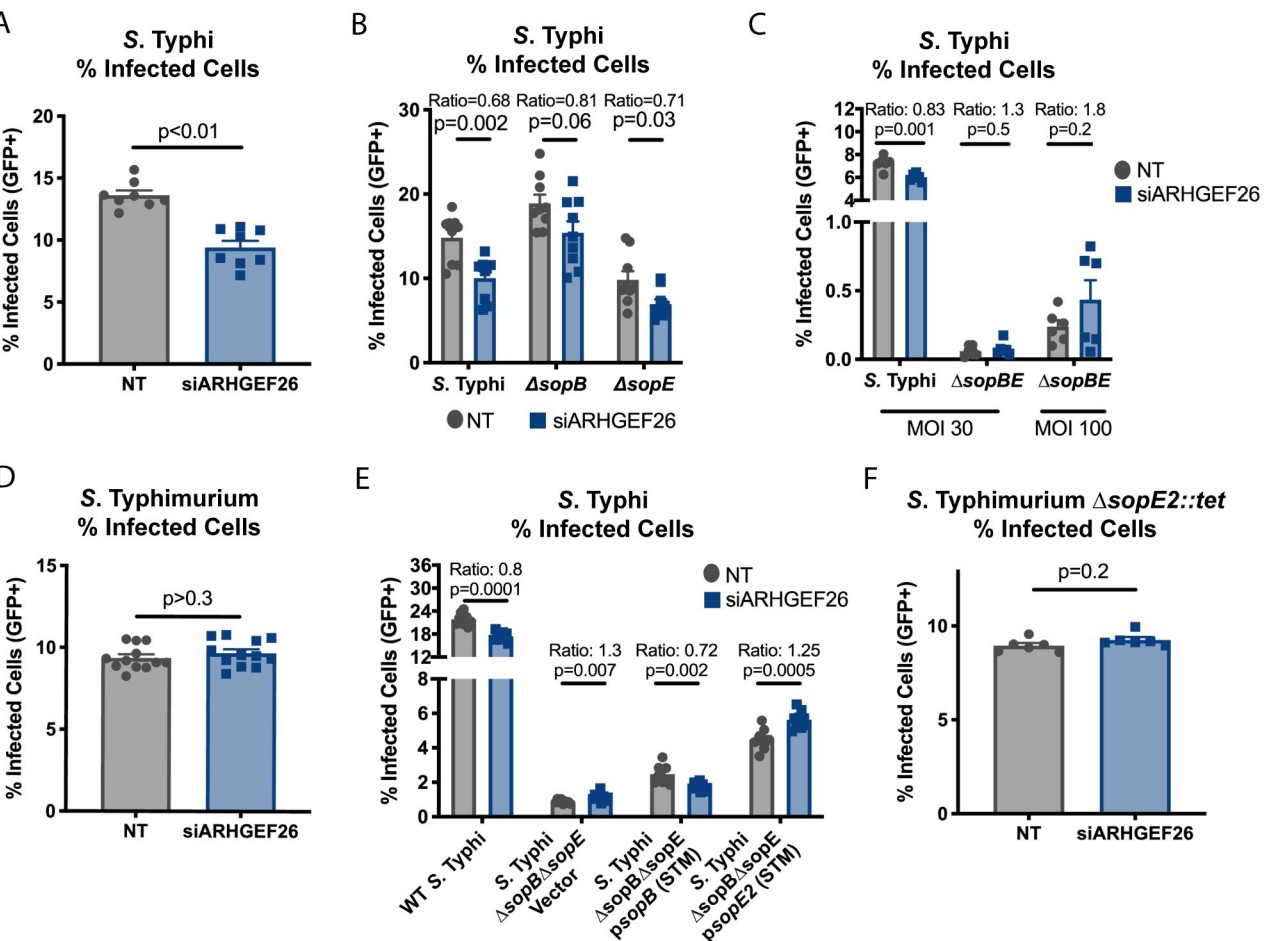

**Fig 3. *ARHGEF26* knockdown reduces HeLa cell susceptibility to *S.* Typhi, but not *S.* Typhimurium, invasion.** (A) RNAi knockdown of *ARHGEF26* in HeLa cells results in reduced susceptibility to *S.* Typhi invasion compared to non-targeting (NT) siRNA. (B, C) The effect of *ARHGEF26* knockdown on *S.* Typhi invasion does not require either *sopB* or *sopE* (B) but does require the presence of at least one effector (C). (D) RNAi knockdown of *ARHGEF26* in HeLa cells does not reduce susceptibility to *S.* Typhimurium invasion. (E) Complementing Δ*sopB*Δ*sopE S.* Typhi with *S.* Typhimruium (STM) *sopB*, but not *S.* Typhimurium (STM) *sopE2* restores the ARHGEF26-dependent invasion phenotype. *sopB* and *sopE2* are ectopically expressed from the pACYC184 plasmid backbone [42]. (F) Knocking out *sopE2* does not render *S.* Typhimurium susceptible to *ARHGEF26*, as a *S.* Typhimurium strain where the gene is replaced with a tetracycline resistance allele (*tet*) also shows no effect. All comparisons are made to transfection with non-targeting (NT) siRNA. Cells were infected at MOI 30 (*S.* Typhi) or MOI 1 (*S.* Typhimurium) for 30 minutes before gentamicin treatment. For panel C, MOI 100 was also used to overcome the significant invasion defect of Δ*sopB*Δ*sopE* and improve infection quantification. For all panels, the percent of host cells infected with *Salmonella* was measured three hours post infection by flow cytometry. All dots represent biological replicates from at least two experiments. *ARHGEF26* knockdown resulted in ~75% reduced transcript abundance (S1F Fig). All ratios are the siARHGEF26 mean divided by the NT mean. Data across experiments were normalized to the grand mean prior to plotting or performing statistics. Bars represent the mean and error bars represent standard error of the mean. P-values for all panels generated by unpaired t-test.

## *DLG1* and *SCRIB*, but not *RHOG* knockdown, phenocopy *ARHGEF26* knockdown

We next tested whether the effects of *ARHGEF26* knockdown could be phenocopied by knocking down *RHOG*. Notably while Patel *et al.* showed that RHOG contributes to SopB-mediated membrane ruffling [7] and invasion [25], more recent reports have demonstrated that RHOG is dispensable for *Salmonella* invasion into fibroblasts [20] and Henle cells [19,20]. In line with this, we found that *RHOG* knockdown (~85% reduction, S1F Fig) did not reduce HeLa susceptibility to *Salmonella* Typhi (Fig 4A) or Typhimurium (Fig 4B) invasion. Curiously, we found that *RHOG* knockdown actually subtly increased invasion.

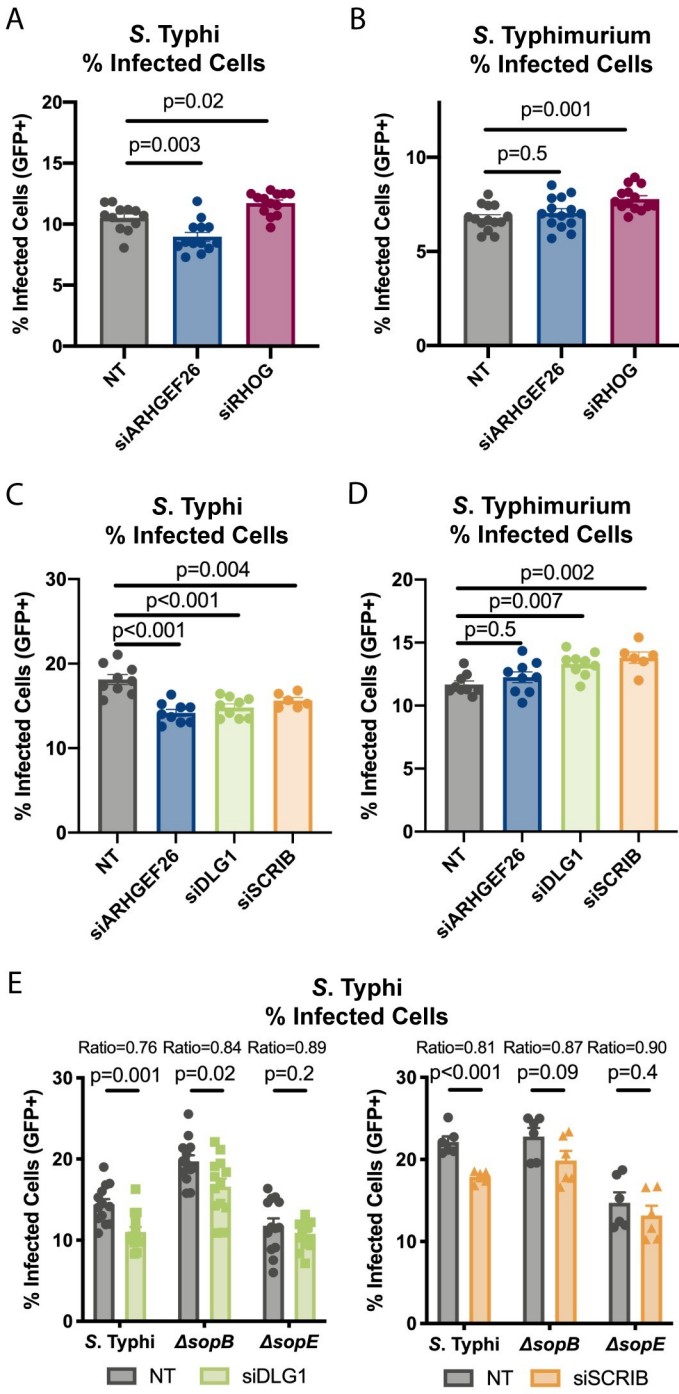

**Fig 4. ARHGEF26 interactors DLG1 and SCRIB, but not RHOG, contribute to HeLa cell susceptibility to *S*. Typhi invasion.** (A,B) *RHOG* knockdown does not reduce *S*. Typhi or *S*. Typhimurium invasion. (C,D) RNAi knockdown of *DLG1* and *SCRIB* in HeLa cells phenocopy the reduction in susceptibility to *S*. Typhi (C), but not *S*. Typhimurium (D) invasion that we observe with *ARHGEF26* knockdown. (E) *DLG1* and *SCRIB* knockdown significantly reduces susceptibility to wild-type *S*. Typhi and Δ*sopB* *S*. Typhi invasion, but only modestly Δ*sopE* invasion. Cells were infected at MOI 30 (*S*. Typhi) or MOI 1 (*S*. Typhimurium) for 30 minutes. Percent infected cells was measured three hours post infection by flow cytometry. All dots represent biological replicates from at least three experiments. *DLG1* knockdown resulted in ~85% transcript reduction, *SCRIB* knockdown resulted in ~90% transcript reduction (S1F Fig). Ratios in E represent the mean of percent infected cells following siRNA treatment divided by the mean percent infected cells following non-targeting (NT) siRNA treatment. Data across experiments were normalized to the grand mean prior to plotting or performing statistics. Bars represent the mean and error bars represent standard error of the mean. P-values

were generated by unpaired one-way ANOVA with Dunnett's multiple comparison test for A-D. For panel E p-values were generated by unpaired t tests.

Following our finding that RHOG did not phenocopy *ARHGEF26* knockdown, we examined whether other canonical ARHGEF26-interacting proteins may contribute to ARHGEF26's role in invasion susceptibility. Previous work has demonstrated that the Scribble complex members DLG1 and/or SCRIB play key roles facilitating ARHGEF26 activity and RHOG activation during human papillomavirus infection [43], as well as in regulating epithelial junction formation, contractability, and lumen formation in 3D cysts [44]. With this in mind, we investigated whether DLG1 and/or SCRIB have roles in *Salmonella* invasion.

Supporting our hypothesis that DLG1, SCRIB, and ARHGEF26 may act through the same pathway, our findings with *DLG1* and *SCRIB* knockdown (*DLG1*: 85% reduced, *SCRIB*: 90% reduced, S1F Fig) broadly phenocopy our results with *ARHGEF26* knockdown. We observed that knockdown of *DLG1* and *SCRIB* each reduced the proportion of HeLa cells invaded by *S.* Typhi (Fig 4C), while just as with *ARHGEF26* knockdown, *DLG1* and *SCRIB* knockdown did not impact the proportion of HeLa cells with *S.* Typhimurium uptake (Fig 4D). Further, *DLG1* and *SCRIB* knockdown reduced Δ*sopB* and Δ*sopE S.* Typhi invasion, though reduced signal to noise ratios measured during Δ*sopE* invasion caused the differences to be statistically insignificant (Fig 4E). Together, these data demonstrate that *DLG1* and *SCRIB* knockdown phenocopy *ARHGEF26* knockdown, thus providing evidence that ARHGEF26-mediated invasion susceptibility may involve interactions with DLG1 and SCRIB, but not RHOG.

## ARHGEF26 requires GEF catalytic activity to increase HeLa cell susceptibility to *S.* Typhi uptake during overexpression conditions

We hypothesize that there are three non-mutually exclusive reasons why *DLG1* and *SCRIB*, but not *RHOG* knockdown phenocopy *ARHGEF26* knockdown. First, it is possible that 85% *RHOG* knockdown (S1F Fig) may not be sufficient to impair RHOG's cellular functions. Alternatively, DLG1 and SCRIB may enable ARHGEF26-mediated activation of other small GTPases. For instance, ARHGEF26 can weakly activate CDC42 and RAC1 *in vitro* [45,46], and could be involved in activating other small-GTPases, such as RHOJ, that impact invasion [19]. Our third hypothesis is that ARHGEF26 may have impacts on *Salmonella* invasion independent of its GEF activity, as has been recently described for ARHGEF26, DLG1, and SCRIB-mediated actomyosin contractility [44]. To determine whether ARHGEF26-mediated invasion susceptibility requires catalytic GEF activity, we assessed which ARHGEF26 domains enhance *S.* Typhi invasion by structure-function analysis (Fig 5A).

Overexpression of ARHGEF26, but not overexpression of a catalytically dead (CD) mutant (E446A, N621A), increased HeLa cell susceptibility to *S.* Typhi invasion (Fig 5B). Additionally, overexpression of mutants lacking the pleckstrin homology (PH) and/or the catalytic Dbl homology (DH) domain failed to promote invasion. All other domains were independently dispensable. While the DH and PH domains were necessary, they were not sufficient, as a DH-PH construct failed to increase the proportion of *S.* Typhi infected cells.

As catalytically active *ARHGEF26* overexpression induces spontaneous membrane ruffling and macropinocytosis, we next examined whether membrane ruffling and invasion could be decoupled. To do this, we overexpressed the *ARHGEF26* constructs and quantified the passive uptake of Δ*prgH S.* Typhi, which cannot dock to host cells or induce invasion (Fig 5C), as well as the number of ARHGEF26[+] membrane ruffles present (Fig 5D and 5E). Across five of our six *ARHGEF26* mutants, we found strong correlations between increased susceptibility to *S.*

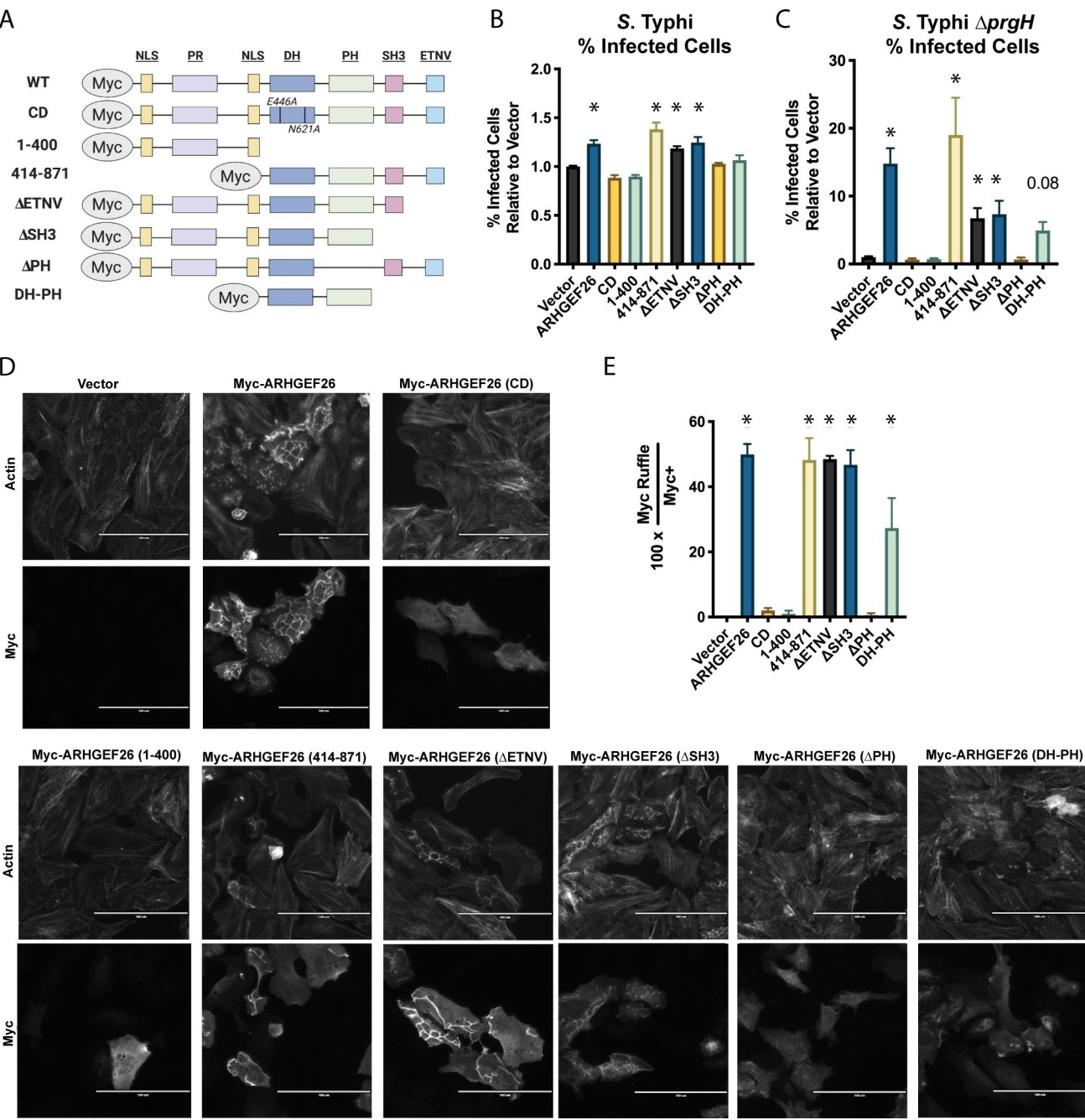

**Fig 5. ARHGEF26 DH and PH domains are required for ARHEGEF26 to induce membrane ruffling and promote *Salmonella* HeLa cell invasion.** (A) Schematic of overexpression constructs used. (B,C) Overexpression of *ARHGEF26* constructs in HeLa cells results in increased wild-type *S*. Typhi infection (B), as well as infection by a SPI-1 secretion mutant (Δ*prgH*) *S*. Typhi (C). Infections were performed at MOI 30 for 60 minutes before gentamicin treatment and quantified 3 hours post infection by flow cytometry. % Infected Cells is reported relative to vector and includes data from at least three independent experiments with three replicates per experiment. Asterisk (*) represents a corrected p-value < 0.05 by Kruskal-Wallis test with Dunn's multiple comparison test. (D) *ARHGEF26* constructs able to induce invasion also induce membrane ruffling. Overexpression of catalytically active constructs in HeLa cells results in membrane ruffling that can be observed both with ARHGEF26 staining using the Myc Tag, as well as by using phalloidin staining to observe actin. (E) Quantification of membrane ruffling confirms correlation with *ARHGEF26*-mediated invasion. Frequency of membrane ruffles was quantified as a percent of Myc+ cells that had a clear Myc+ ruffle. Ruffle abundance was quantified from four independent experiments. Values were generated by quantifying ruffles from five separate fields of view. The presence of a ruffle was confirmed by examining phalloidin stained actin at that site. Scale bar is 100 μM. Asterisk (*) represents a corrected p-value < 0.05 generated by one-way ANOVA of the log(X+1) transformed values with Dunnett's multiple comparisons test.

Typhi invasion (Fig 5B), Δ*prgH S*. Typhi uptake (Fig 5C), and membrane ruffling (Fig 5D and 5E). The one exception was our DH-PH mutant, which was not able to promote wild-type *S*. Typhi uptake but did modestly increase Δ*prgH S*. Typhi invasion (p = 0.08) and induce small membrane ruffles. This indicates that the small ruffles are sufficient to drive Δ*prgH* uptake but are an insignificant addition to the typical *S*. Typhi-induced membrane ruffles. Overall, these data suggest that *ARHGEF26* overexpression affects invasion into HeLa cells by increasing membrane ruffling and macropinocytosis.

We conclude that, under overexpression conditions, *ARHGEF26* constructs that do not show catalytic activity but are still able to bind DLG1 and SCRIB, such as the 1–400 construct [44], are not able to drive *Salmonella* invasion. Thus, our RNAi and overexpression results support a model where interaction of DLG1/SCRIB, nucleotide exchange, and membrane ruffling contribute to ARHGEF26's role in promoting HeLa cell susceptibility to invasion, despite being RHOG independent.

## ARHGEF26 does not show significant phosphoinositide binding using a dot blot assay

While our data suggest that DLG1 and SCRIB may contribute to ARHGEF26 activity, the prevailing model postulates that ARHGEF26 is guided to the plasma membrane through its pleckstrin homology (PH) domain [7], which in some proteins can bind phosphoinositides [47]. Under this model, ARHGEF26 localization is regulated by SopB's effects on phosphoinositides [6,7,9,21,48–50]. Supporting this, our data demonstrates that the ARHGEF26 PH domain is required to increase invasion (Fig 5). However, many PH domains either completely lack canonical phosphoinositide binding or have phosphoinositide binding that is physiologically irrelevant [47,51]. Therefore, PH-dependence alone is insufficient to implicate SopB-generated phosphoinositides as this could suggest either that ARHGEF26 binds phosphoinositides in order to function, or simply, that this mutation disrupts the catalytic domain as has been shown for other RHO GEFs [46].

To directly test whether ARHGEF26 binds phosphoinositides, we performed a dot blot assay in which different phosphoinositide species are dotted on a membrane and exposed to ARHGEF26. Across a variety of conditions—including different ARHGEF26 constructs, cellular sources of protein, and blocking solutions—we did not detect strong phosphoinositide binding (S2 Fig). We also tested whether co-expression of ARHGEF26 and RHOG could drive phosphoinositide binding, as occurs with the RHOG GEF Trio [52], but did not observe increased signal. Under some conditions, weak and non-specific signal appeared on the dots of some phosphoinositide species, but this was independent of the PH domain and difficult to distinguish from background noise. This contrasted with our positive control, the AKT-PH domain, which demonstrated robust and highly specific binding. By considering the difference in signal between the ARHGEF26 constructs and AKT-PH domain, as well as the history of this assay exaggerating the affinity for proteins with phosphoinositides [53], we surmise that even if this signal is the result of a weak affinity for phosphoinositides, this affinity is unlikely to play any physiological role *in vivo*. Therefore, while we cannot firmly rule out that ARHGEF26 binds phosphoinositides using this assay, these data do not support PH domain-mediated phosphoinositide binding directing ARHGEF26 localization.

Together, our results suggest a new model for the role of *ARHGEF26* during *S*. Typhi HeLa cell invasion. We propose that ARHGEF26 potentially stimulates invasion independently of RHOG and that recruitment of ARHGEF26 to the site of invasion is not dependent on SopB-mediated phosphoinositide changes. Instead, our results demonstrate the SCRIB-DLG1-ARHGEF26 complex is important for invasion even with SopE stimulation being the primary route of invasion.

## ARHGEF26 and DLG1 promote susceptibility of polarized MDCK cells to *S.* Typhimurium, but not *S.* Typhi, invasion

Recent work has demonstrated that while a convenient model, HeLa cells and other unpolarized epithelial cell lines do not perfectly recapitulate interactions between *Salmonella* and gut epithelia [54]. Therefore, we sought to understand how ARHGEF26 contributes to *Salmonella* invasion of polarized MDCK cells expressing an *ARHGEF26* targeting shRNA [44]. Interestingly, we found *ARHGEF26* knockdown cells were significantly more resistant to *S.* Typhimurium invasion compared to wild-type cells, but not *S.* Typhi invasion (Fig 6A), a striking contrast to our results in HeLa cells (Fig 3). Of note, this is the largest phenotype in invasion susceptibility we observed across our three cell lines, suggesting that while ARHGEF26 may be an accessory to *Salmonella* invasion in LCLs and HeLa cells, it is a critical component of the invasion machinery in polarized MDCK cells. In line with previous reports, a significant fraction of the *S.* Typhimurium uptake into polarized epithelia we observed was independent of the effectors SopB and SopE2 [54], but, surprisingly, the effects of *ARHGEF26* knockdown on invasion susceptibility were entirely independent of these effectors as well (Fig 6A). We next examined whether these effects could be phenocopied by perturbing *DLG1* by generating a *DLG1* MDCK knockout line (S3 Fig), and indeed, *DLG1* knockout cells were resistant to *S.* Typhimurium invasion in a SopB- SopE2-independent manner (Fig 6B).

In order to gain insight into how ARHGEF26 influences susceptibility to *S.* Typhimurium invasion, we performed confocal microscopy to examine how many bacteria successfully invaded wild-type or shARHGEF26 expressing polarized MDCK cells. For this technique, invasion was allowed to proceed for 30 minutes prior to the addition of gentamicin and IPTG, and invasion was quantified two hours post infection to minimize measuring effects on early bacterial replication. Unlike our flow cytometric method which considers the percentage of host cells invaded, this method allows us to examine invasion from the bacterial perspective and analyze both the frequency and distribution of invasion throughout the well. Strikingly, not only did *shARHGEF26* expressing cells have ~70% relative reduction in the percentage of cells invaded by *S.* Typhimurium (Fig 6A), but the amount of total invasion was almost completely ablated in these cells (Fig 6C and 6D). This difference appears to be driven by cooperative invasion of both the same cell and neighboring cells, as wild-type cells had dense clusters of invaded *S.* Typhimurium, while *shARHGEF26* cells largely have one to two bacteria within each cluster (Fig 6D). While we cannot rule out that some replication occurred, we consider it unlikely given the early timepoint. Curiously, while we observed a smaller proportion of cells were infected with *ΔsopBΔsopE2 S.* Typhimurium (Fig 6A), we actually observed a modest increase in the total amount of enumerated bacteria within these cells (Fig 6C and 6D), though again, *ARHGEF26* knockdown almost completely ablated this. No differences were observed in *S.* Typhi invasion, though very little *S.* Typhi invasion was observed at all. Together these data demonstrate that ARHGEF26 and DLG1 are critical for *S.* Typhimurium invasion into polarized epithelia and suggest that these two proteins may cooperatively work together to facilitate uptake.

Our comparative analysis of invasion in three cell types has demonstrated three distinct invasion susceptibility phenotypes: (1) ARHGEF26 contributes modestly to susceptibility of LCLs to *S.* Typhi and *S.* Typhimurium invasion, (2) ARHGEF26 contributes to HeLa susceptibility to SopB- and SopE-mediated *S.* Typhi invasion, and (3) ARHGEF26 is a critical determinant of polarized MDCK susceptibility to SopB- and SopE2-independent *S.* Typhimurium invasion. This highlights that there is clearly unappreciated complexity not only in how ARHGEF26 is manipulated and utilized by *Salmonella*, but in how invasion broadly differs according to host by serovar interactions.

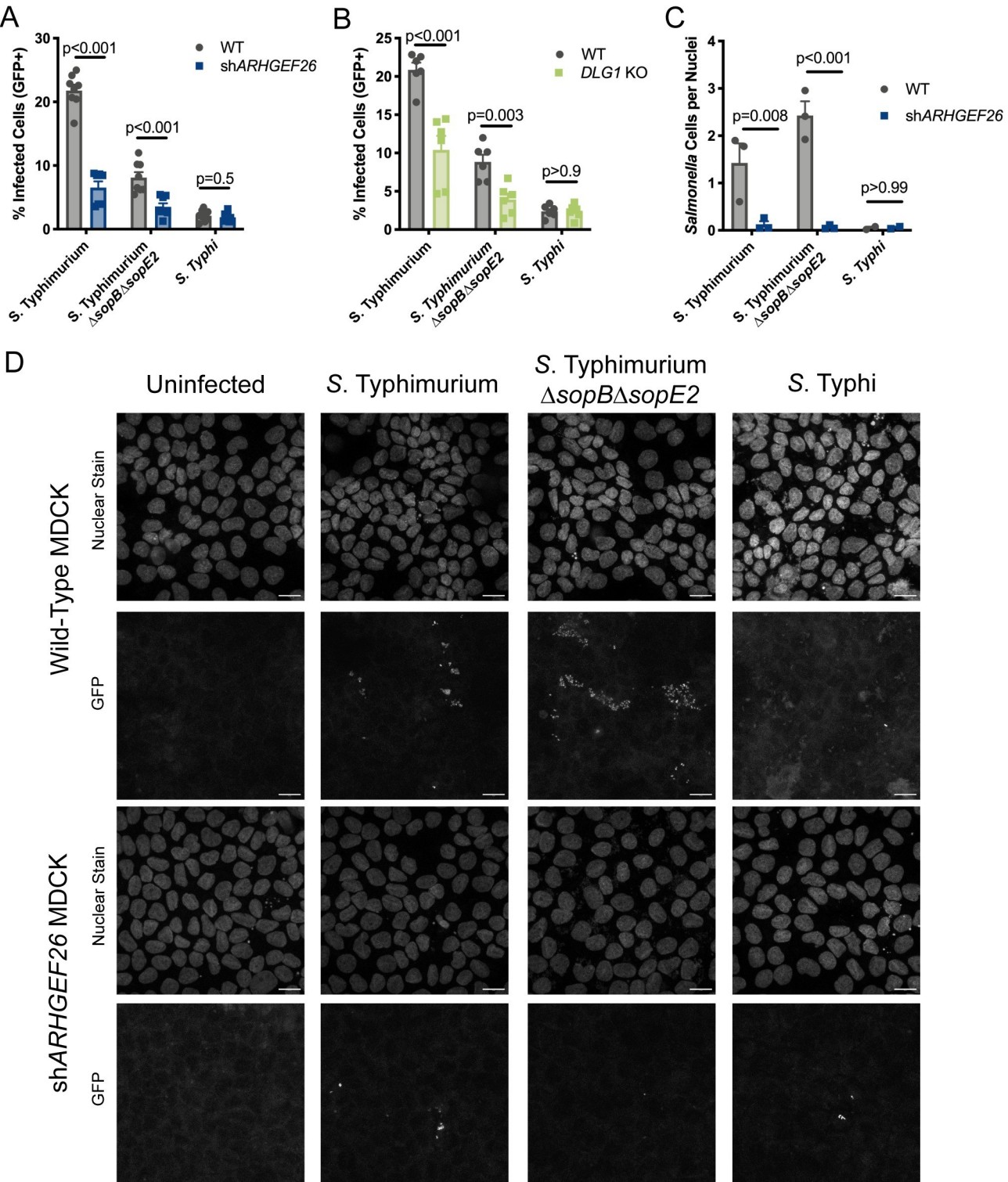

**Fig 6. *ARHGEF26* knockdown reduces polarized MDCK cell susceptibility to *Salmonella* Typhimurium invasion.** (A) *shARHGEF26* expressing polarized MDCK cells have reduced susceptibility to *S.* Typhimurium invasion independent of SopB and SopE2 compared to wild-type (WT) polarized MDCKs. (B) *DLG1* knockout polarized MDCK cells have reduced susceptibility to *S.* Typhimurium invasion independent of SopB and SopE2. For panels A and B, cells were infected at MOI 30 (*S.* Typhimurium) or MOI 100 (*S.* Typhi) for 30 minutes before gentamicin treatment. The percent of host cells infected with *Salmonella* was measured three hours post infection by flow cytometry. All dots represent biological replicates from at least three experiments normalized to the grand mean. Bars represent the mean and error bars represent standard error of the mean. P-values for panels generated

by unpaired t-test. (C, D) *shARHGEF26* ablates invasion of polarized MDCK cells by *S*. Typhimurium. Invasion measured by quantifying the number of GFP+ bacteria two hours after infection, following 30 minutes of invasion as described above. For panel C, each dot represents the number of bacteria divided by the number of nuclei observed within a randomly selected frame, with one frame per experiment and at least two experiments performed per condition. At least 70 nuclei were present within each frame for quantification. P-values calculated by unpaired t-test. Panel D is a representative experiment from C, and the scale bar is 16μm. For all experiment, cells were polarized by plating at high density on untreated glass chamber well slides and incubating for four days with media changes every 48 hours.

## ARHGEF26 is required for *S*. Typhimurium virulence during an enteric fever model of infection

Following these results, it was crucial to define ARHGEF26's role in *Salmonella* pathogenesis in mice. As *S*. Typhi is a human-specific pathogen, we focused specifically on how ARHGEF26 influences murine *S*. Typhimurium pathogenesis. Based on the data from H2P2 and *ARHGEF26* knockdown in LCL and MDCK cells, we hypothesized that ARHGEF26 is required for establishment of *S*. Typhimurium in the mammalian gut. To test this hypothesis, we utilized *Arhgef26*<sup>-/-</sup> C57BL/6J mice [55] to assess the ability of *S*. Typhimurium to establish infection using the oral enteric fever model of infection (Fig 7A). In this model, SPI-1 secretion is required for *S*. Typhimurium colonization and persistence in the mammalian ileum and helps facilitate *S*. Typhimurium dissemination to the spleen [3]. In support of our hypothesis that *Arhgef26* promotes invasion *in vivo*, *Arhgef26*<sup>-/-</sup> mice showed significantly lower *S*. Typhimurium burdens in the ileum, but this effect was considerably smaller in the spleen (Fig 7B).

As *Arhgef26*<sup>-/-</sup> mice phenocopy the effects of SPI-1 knockout on ileal burden, we next examined whether the effect of *Arhgef26* knockout requires SPI-1. To genetically test this hypothesis, wild-type and *Arhgef26*<sup>-/-</sup> mice were infected with wild-type, Δ*prgH*, and Δ*sopB S*. Typhimurium. Supporting that a functional SPI-1 secretion system is required for ARHGEF26 to affect *S*. Typhimurium fitness, the difference in burden between wild-type and *Arhgef26*<sup>-/-</sup> mice was significantly reduced when mice were infected with Δ*prgH* bacteria (Fig 7C). In contrast, differences between wild-type and *Arhgef26*<sup>-/-</sup> mice were slightly reduced but broadly maintained when infected with a Δ*sopB S*. Typhimurium strain (Fig 7C), once again demonstrating that the previous model that ARHGEF26 contributes specifically to SopB-mediated fitness is incomplete.

## SopB- and SopE2-mediated *S*. Typhimurium fitness depends completely on ARHGEF26 in the murine gut

We next examined whether the canonical ruffle-inducing SPI-1 effectors, SopB and SopE2, were together required for the impacts of *Arhgef26* knockout. Strikingly, the entire difference between wild-type and *Arhgef26*<sup>-/-</sup> was ablated if mice were infected with Δ*sopB*Δ*sopE2 S*. Typhimurium (Fig 7D). This suggests that the differences in burden between wild-type and knockout mice in the enteric fever model of infection is entirely dependent on SopB- SopE2--mediated processes, and not by alternative methods of *S*. Typhimurium invasion that occur *in vivo* [54]. However, perhaps the most shocking revelation from this experiment was that ARHGEF26 is, in turn, required for the effects of SopB and SopE2 on bacterial burden (Fig 7D). In C57BL/6J mice, we observed a substantial decrease in bacterial burden with mice infected with Δ*sopB*Δ*sopE2* bacteria compared to those infected with wild-type *S*. Typhimurium, with the number of Δ*sopB*Δ*sopE2* bacteria recovered roughly equal to the number of CFU/g we observed in *Arhgef26*<sup>-/-</sup> mice infected with wild-type *S*. Typhimurium. However, in *Arhgef26*<sup>-/-</sup> we observed no such difference in comparing CFUs of wild-type vs. Δ*sopB*Δ*sopE2 S*. Typhimurium (p = 0.9). Thus, we conclude that SopB- SopE2-mediated survival of *S*. Typhimurium in the murine ileum is entirely ARHGEF26 dependent.

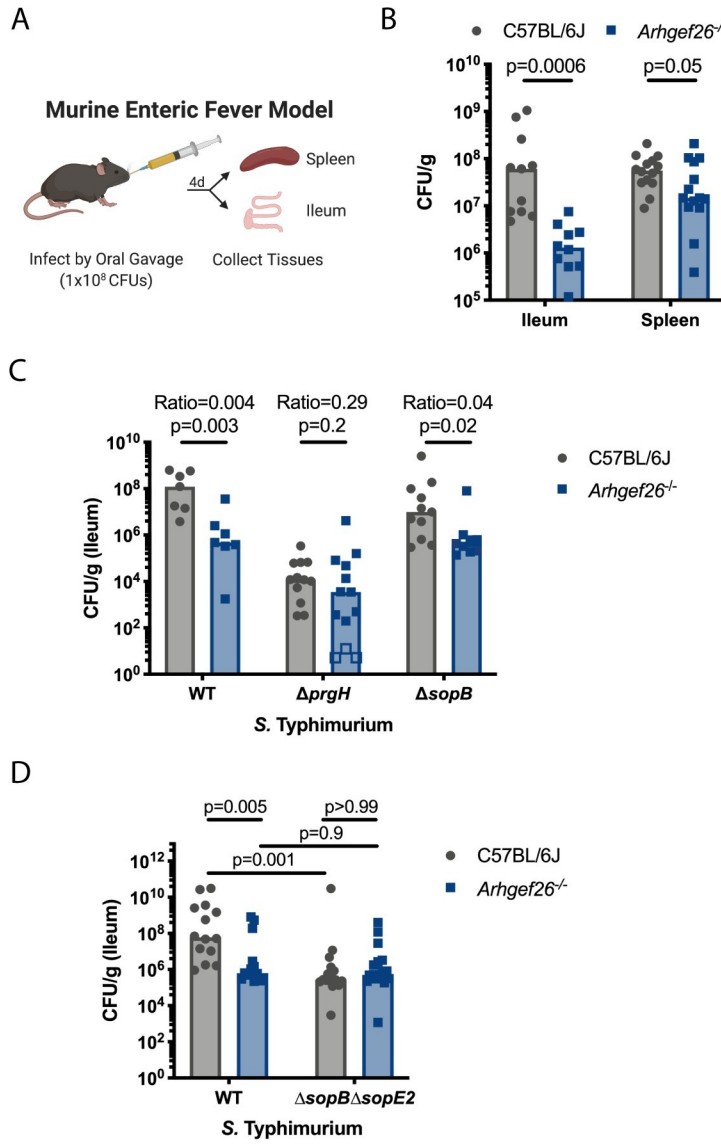

**Fig 7. *Arhgef26* is critical during *S.* Typhimurium infection in the enteric fever murine model.** (A) Schematic of the murine enteric fever infection model. (B) *Arhgef26*[-/-] mice have reduced ileal and splenic bacterial burden in the enteric fever infection model compared to C57BL/6J mice. P-values generated by Welch's t-test. (C) Effects of *Arhgef26* knockout on bacterial fitness depends on *prgH*, but not *sopB*. Open boxes represent mice where no colony forming units (CFU) were recovered from *ARHGEF26*[-/-] mice and the CFU/g was set to the limit of detection. P-values were generated by unpaired t-tests on log transformed values. (D) ARHGEF26 requires SopB and SopE2 to impact ileal burden, and SopB and SopE2 require ARHGEF26. P-values from one-way ANOVA with Sidak's multiple comparison test on log transformed values. Ratios in C represent median CFU/g recovered from *Arhgef26*[-/-] mice divided by the median CFU/g recovered from C57BL/6J mice. For all experiments, mice were infected with 1x10[8] *S.* Typhimurium CFU and tissues were harvested four days post infection for CFU quantification. All bar graphs contain data generated from at least two experiments, and each dot represents a single mouse. Data for B and C were normalized to the grand mean prior to plotting or performing statistics. Bars represent the median. All mice were age and sex matched within experiments, with both sexes represented in all experiments.

## ARHGEF26 contributes to *Salmonella*-induced inflammation in HeLa cells

In addition to enabling *Salmonella* invasion, interactions between the SPI-1 secreted effectors and host machinery drive inflammation that is characteristic of *Salmonella* infections

(Reviewed [56]). For instance, during *Salmonella* invasion, SopB and SopE activate CDC42, which goes on to enable the formation of a PAK1–TRAF6–TAK1 complex, NF-κB activation, and increased IL-8 production [7,57–61]. Other work has suggested NOD2 and RIPK2 also contribute to CDC42- and RAC1-mediated inflammation [62]. Further, studies using dominant negative constructs have suggested that CDC42 and Rac1 are required for secretion of IL-8 from polarized cells *in vitro*, presumably through changes to the cytoskeleton [63]. Based on our findings that ARHGEF26 is a SPI-1 manipulated and critical GEF during *Salmonella* invasion, we hypothesized that it may also have a role in mediating *Salmonella*-induced inflammation.

To examine if ARHGEF26 contributes to inflammation, we knocked down *ARHGEF26* and *RHOG* in HeLa cells and measured IL-8 abundance in supernatant. In supernatant from uninfected cells, IL-8 was reduced following *ARHGEF26* or, interestingly, *RHOG* knockdown (Fig 8A). This suggests that even under basal conditions, ARHGEF26 regulates inflammation, potentially through interactions with RHOG.

Overexpression experiments confirmed the importance of ARHGEF26 and RHOG in regulating basal inflammatory cytokine production. *ARHGEF26* overexpression resulted in significantly increased IL-8 production (Fig 8B). Surprisingly, a partial effect was observed with overexpression of the catalytically dead construct (Fig 8B), demonstrating that ARHGEF26 does not require GEF activity to influence cytokine production. Overexpression of wild-type *RHOG* did not increase IL-8 in supernatant, but overexpression of a constitutively active *RHOG* construct (Q61L) resulted in very robust cytokine production (Fig 8B). Together, our knockdown and overexpression data demonstrate that ARHGEF26 promotes inflammatory cytokine production and involves both GEF-dependent and GEF-independent mechanisms.

We next sought to examine whether ARHGEF26 regulates inflammation during *Salmonella* infection. Infection with *S.* Typhi caused induction of IL-8, and levels were moderately lower in supernatants with *ARHGEF26*, but not with *RHOG* knockdown (Fig 8C). This aligned with what we observed with invasion, suggesting either that related mechanisms could be contributing to ARHGEF26-mediated invasion and inflammation, or that reduced inflammation is driven by reductions in invasion. However, in contrast to our invasion data, where *S.* Typhimurium HeLa cell invasion was not affected by *ARHGEF26*, we found that *ARHGEF26*, but not *RHOG*, knockdown also moderately reduced IL-8 abundance following *S.* Typhimurium infection (Fig 8D). This suggests that *S.* Typhimurium-mediated ARHGEF26-enhanced IL-8 production is invasion-independent. Together, these data demonstrate that ARHGEF26 and RHOG have context specific roles through which they promote inflammation in HeLa cells.

Importantly, the magnitude of the *siARHGEF26*-mediated cytokine reduction for both *S.* Typhi- and *S.* Typhimurium-induced IL-8 was much larger than the reduction seen in uninfected cells (~1,400pg/mL and ~700pg/mL, accordingly vs ~200pg/mL). This substantially larger phenotype with infection leads us to conclude that our results are not simply consequences of lower basal levels of cytokine production following *ARHGEF26* knockdown. Instead, we propose that there are multiple levels of cytokine regulation by ARHGEF26, and that this could have important implications during *Salmonella* infection.

## ARHGEF26 contributes to inflammation during a colitis model of *S.* Typhimurium infection

To test whether ARHGEF26 contributes to *Salmonella*-induced inflammation *in vivo*, we again examined *Arhgef26*$^{-/-}$ mice. Importantly, while the enteric fever model is able to measure SPI-1- and invasion-dependent *Salmonella* fitness *in vivo*, it does not mimic the natural progression of most *S.* Typhimurium illness in humans as it causes very little inflammation. In

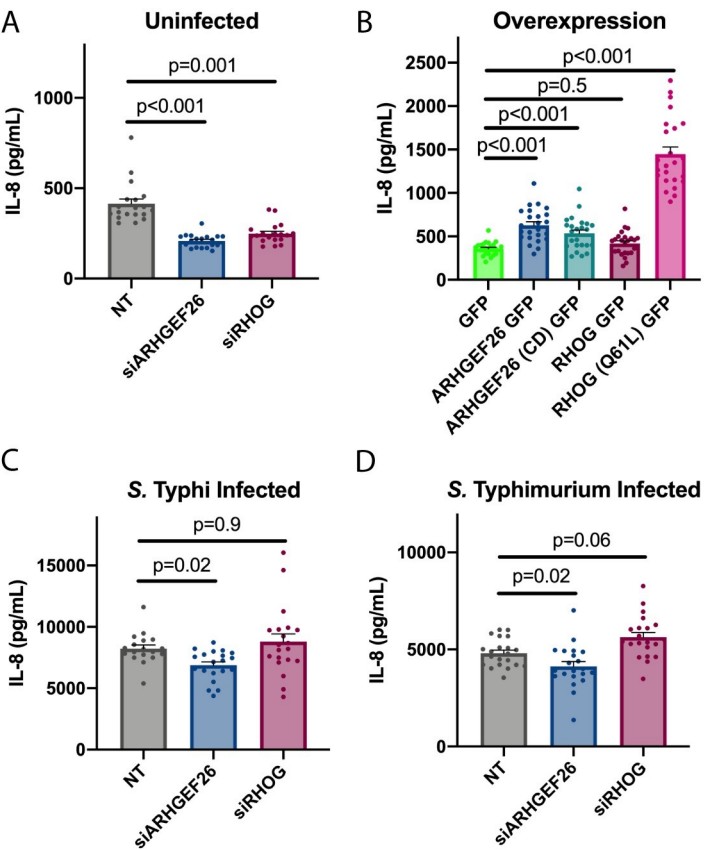

**Fig 8. ARHGEF26 and RHOG are context-dependent enhancers of IL-8 abundance in HeLa cell supernatant.** (A) *ARHGEF26* and *RHOG* knockdown results in less IL-8 secretion into HeLa cell supernatant than non-targeting (NT) siRNA. Media was changed two days post transfection and supernatant was collected 8 hours later. (B) Overexpression of *ARHGEF26* and *RHOG* in HeLa cells increases IL-8 cytokine abundance in supernatant. Media was changed on transfected cells 18–24 hours post transfection and supernatant was collected 6 hours later. (C, D) *ARHGEF26*, but not *RHOG*, knockdown reduces IL-8 abundance in supernatant following *S.* Typhi infection (C) and *S.* Typhimurium infection (D). For C and D, cells were infected two days post transfection. Cells were infected with *S.* Typhi (MOI 30) or *S.* Typhimurium (MOI 1) for one hour before gentamicin addition. Media was changed two hours before infection and supernatant was collected six hours after infection. Cytokine abundance was measured by ELISA. For all graphs, dots represent a single well and data were collected across seven independent experiments. Data were normalized to the grand mean prior to plotting or performing statistics. For (C), two outliers identified by ROUT (Q = 0.1%) were removed from the non-targeting group. These values (17,924 pg/mL and 22,083 pg/mL) inflated the mean of the NT group, making the ARHGEF26 effect size artificially large, and the p-value artificially low (p = 0.002). P-values were calculated using a one-way ANOVA with Dunnett's multiple comparison test on the $\log_2$ transformed data. For all graphs central tendency is the mean, error bars are the standard error of the mean.

order to mimic *S.* Typhimurium colitis disease in murine models, microbiota must be reduced with streptomycin pretreatment prior to infection [11] (Fig 9A). Interestingly, in this model, SPI-1 secretion does not impact bacterial burden at early timepoints but instead is required for inflammation onset [10,11]. These differences mean that data from the colitis model cannot be combined with data from the enteric fever model, but that interpreting the colitis model separately can provide important information about pathogenic processes that do not occur in the enteric fever model.

Using the murine colitis model, we examined whether ARHGEF26 promoted inflammation and pathology by infecting wild-type and *Arhgef26*[-/-] mice following streptomycin pretreatment. Indeed, while there was no effect of *Arhgef26* deletion on the recovery of *S.* Typhimurium CFUs from ileum, cecum, or colon two days post infection (Fig 9B), these sites

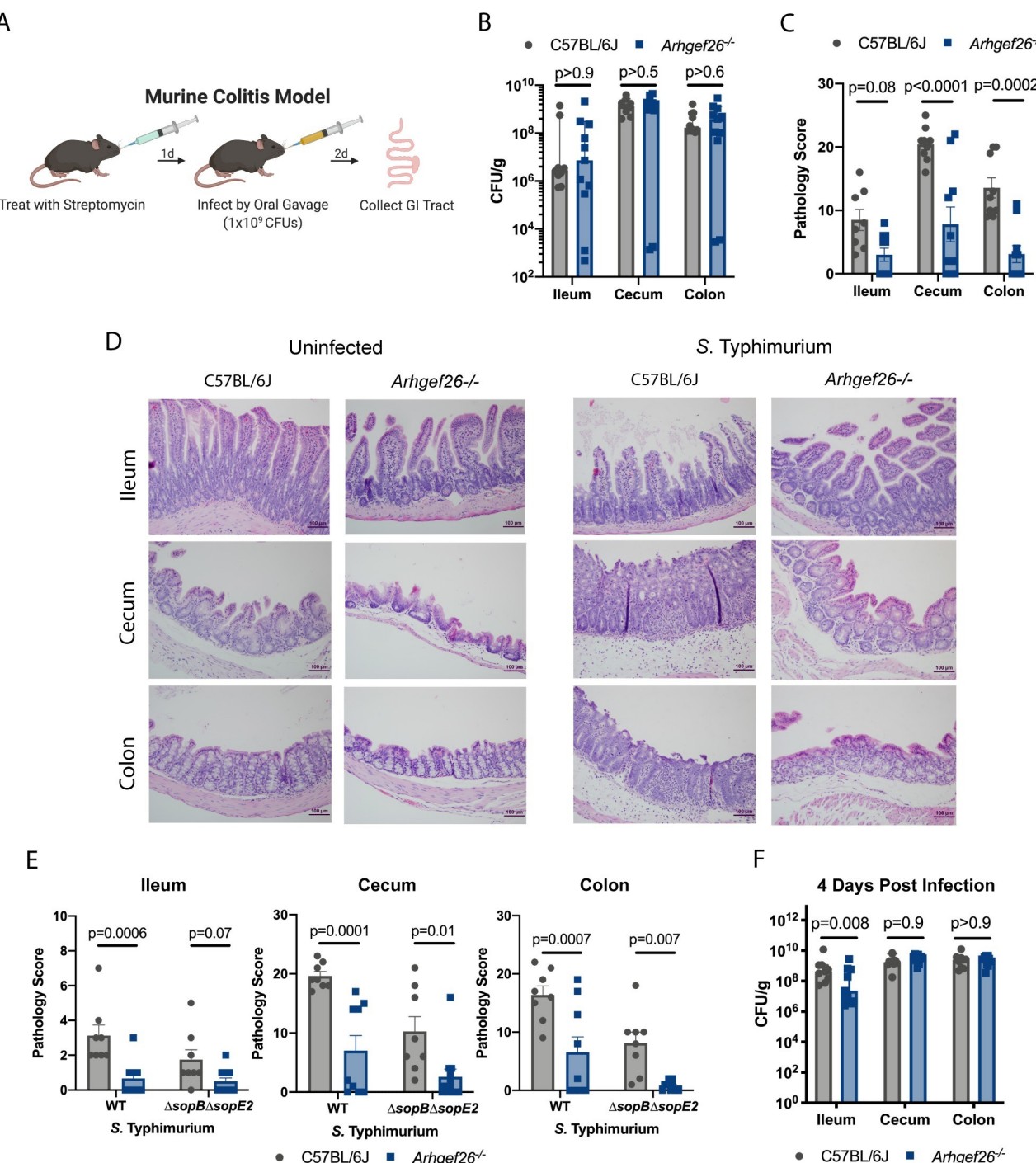

**Fig 9. *Arhgef26* enhances inflammation in mice.** (A) Schematic for murine colitis infection model. Mice are pretreated with streptomycin one day before infection with 1x10$^9$ *S.* Typhimurium CFUs. Tissues are collected 2 days post infection for CFU quantification or histological analysis. (B) *Arhgef26$^{-/-}$* mice have no reductions in gastrointestinal tract bacterial burdens in the colitis infection model two days post infection. p-values generated by two-way ANOVA on the log transformed values with Sidak's multiple comparison test, bars mark the median, error bars the 95% confidence interval. (C) *Arhgef26$^{-/-}$* mice have significantly reduced inflammation following infection compared to C57BL/6J mice in the colitis infection model. Pathology scores were generated in a blinded fashion by a trained pathologist and are broken down in S5 Data. P-values by two-way ANOVA with Sidak's multiple comparison test, the bars mark the mean, and the error bars are the standard error of the mean. (D) Examples of differential pathology following infection between C57BL/6J and *Arhgef26$^{-/-}$* mice. Scale bar is 100 μM. (E) Impact of *Arhgef26* on inflammation does not depend on SopB or SopE2. P-values generated by two-way ANOVA with Sidak's multiple comparison test, bars mark the median, error bars the 95% confidence interval. By two-way ANOVA, no tissue had a significant (p<0.05) interaction term. (F) Differences in inflammation between C57BL/6J mice and *Arhgef26$^{-/-}$* mice do not impact *S.* Typhimurium burdens in the

cecum or colon four days post infection. p-values generated by two-way ANOVA on the log transformed values with Sidak's multiple comparison test, bars mark the median, error bars the 95% confidence interval. For all bar graphs, each dot represents a single mouse from two experiments and all mice were age- and sex-matched, and both sexes were present in all experiments.

demonstrated significantly reduced inflammation-associated pathology in *Arhgef26*$^{-/-}$ mice (Fig 9C and 9D, and S1 Data). This was most striking in the cecum and colon where inflammation is most severe in wild-type mice.

We next examined whether this difference in inflammation depended on the *S.* Typhimurium effectors SopB and SopE2. Consistent with past work, *ΔsopBΔsopE*2 mutants still induced significant amounts of inflammation in the colitis model of infection [10], though less inflammation was observed compared to mice infected with wild-type *S.* Typhimurium (Fig 9E and S1 Data). The differences between wild-type mice and *Arhgef26*$^{-/-}$ mice was not ablated by infecting with the mutant bacteria, though the effect on ileal inflammation was partially reduced. Because of the limited inflammation that occurs in the *Arhgef26*$^{-/-}$ mice paired with the limit of detection of the assay, we cannot definitively conclude that there is no interaction between ARHGEF26 and the SopB/E2 effectors in this model. However, interactions with SopB and SopE2 are clearly not the primary method by which ARHGEF26 contributes to *S.* Typhimurium-induced inflammation.

## ARHGEF26 does not contribute to bacterial burden in the colon or cecum in the murine colitis model

Previous work has demonstrated that *S.* Typhimurium-induced inflammation can aid *Salmonella* fitness in the large intestine through clearance of competing microbiota [64]. Therefore, we hypothesized that the reduced inflammation we observe with *Arhgef26*$^{-/-}$ mice may result in reduced bacterial burden in the cecum and colon at later timepoints. However, we did not observe reduced burden at these sites four days post infection (Fig 9F). We did observe differences in the ileum at this time point, though whether this is due to reduced inflammation or reduced invasion is unclear.

Considering each mouse model separately, we can conclude ARHGEF26 plays a critical, multifaceted role in enabling *S.* Typhimurium to utilize the SPI-1 secretion system to cause disease in both models of murine infection. In the enteric fever model, ARHGEF26 mediates a ~2 log difference in ileal burden that is entirely dependent on sopB and sopE2. In the colitis model, ARHGEF26 has minimal effect on burden but plays a crucial role in inflammation that is largely SopB/SopE2-independent.

## Discussion

Beginning with a candidate pathway approach paired with the Hi-HOST cellular GWAS platform, we identified a QTL in the *ARHGEF26* gene that influences host cell invasion, carried out functional studies that reshape our understanding of how ARHGEF26 stimulates invasion, and revealed a new role for ARHGEF26 in regulating inflammation in cells and mice. Our findings are summarized in Table 1.

One important question moving forward is how the rs993387 locus impacts *ARHGEF26* activity. As noted, published eQTL datasets have not provided a consistent answer for the association of rs993387 with *ARHGEF26* mRNA levels [37,38]. This inconsistency is likely driven by the fact that while *ARHGEF26* expression is detectable based on quantitative PCR (C$_T$ ~30) and RNA-seq datasets in LCLs [38,65], it is a low abundance transcript. Additionally, we have not been able to reliably detect protein levels using either antibodies or mass spectrometric

**Table 1. ARHGEF26 has context-dependent and pleotropic roles during *Salmonella* pathogenesis.**

| | | Impact of *ARHGEF26* Knockdown/ Knockout | | Impact of Additional Bacterial and Host Proteins | |
|---|---|---|---|---|---|
| | | **S. Typhi** | **S. Typhimurium** | **SopB/SopE/SopE2 Dependence** | **Phenocopying Host Proteins** |
| **Invasion Susceptibility (Relative to Non-Targeting or WT)** | **Lymphoblastoid Cells** | Reduced by ~15% | Reduced by ~10% | NT | NT |
| | **HeLa Cells** | Reduced by ~30% | No Effect | Partially SopB/SopE dependent | RHOG does not phenocopy, DLG1 and SCRIB phenocopy |
| | **Polarized MDCK** | No Effect | Reduced by ~70% | SopB/SopE2 independent | DLG1 phenocopies |
| **IL-8 Secretion (Relative to Non-Targeting)** | **Uninfected HeLa Cells** (Reduced by ~200pg/mL) | N/A | N/A | N/A | RHOG phenocopies |
| | **Infected HeLa Cells** | Reduced by ~1,400 pg/mL | Reduced by ~700pg/mL | NT | RHOG does not phenocopy |
| **Mouse Models (Relative to C57BL/6J)** | **Enteric Fever** | N/A | Ileal CFU/g reduced by 2 log | Fully SopB/SopE2 dependent | NT |
| | **Colitis** | N/A | Reduced intestinal pathology scores | SopB/SopE2 independent | NT |

N/A–Not Applicable; NT–Not Tested

approaches. Of note, there is an ARHGEF26 anti-sense transcript (*ARHGEF26-AS1)*, for which rs993387 associates with expression in some tissues (most strongly in tibial nerve p = 6.9x10$^{-22}$) [37]. Also, rs993387 is reported as an *ARHGEF26* splicing QTL in both sigmoid (p = 1.1x10$^{-9}$) and transverse colon (p = 5.4x10$^{-5}$) [37,38]. Thus, though there are several plausible mechanisms, we do not know if rs993387 (or a causal variant in LD) affects ARHGEF26 mRNA levels, splicing, protein levels, or protein function, and such experiments have been technically challenging.

A second unanswered question is how natural variation regulating *ARHGEF26* could impact LCL invasion susceptibility. Small changes in *ARHGEF26* expression or function could change the rate of membrane ruffling as we observed in the HeLa cell knockdown and overexpression systems. Alternatively, changes in *ARHGEF26* could impact the size of membrane ruffles and the efficiency with which macropinocytosis occurs at the site of invasion. A comparable phenomenon is observed with our overexpression system, as constructs with low activity (*e.g.* the DH-PH construct) form small, contained ruffles (Fig 5D). In contrast, other constructs (*e.g.* the 414–871 construct) form large ruffles that were able to increase *S.* Typhi and Δ*prgH S.* Typhi invasion to levels even above what we observe with wild-type *ARHGEF26* overexpression (Fig 5B and 5C). Changing the size of membrane ruffles could have impacts beyond simply enabling the ARHGEF26-recruiting bacteria to invade the cell, as previous work has demonstrated that *Salmonella* swimming at the cell surface use membrane ruffling induced by other bacteria as a signal to begin their own invasion, thus engaging in a sort of cooperative behavior [66]. Thus, large ARHGEF26-induced membrane ruffles could serve as a mechanism for (a) efficient macropinocytosis, and (b) cooperative host cell invasion. An additional consideration is that while our results in LCLs and HeLa cells are likely driven by SopB- and SopE/E2-driven membrane ruffling, our results in MDCK cells are SopB and SopE2 independent. Thus, ARHGEF26 also may regulate the recently described [54], and not yet fully understood, discreet-invasion pathway. How ARHGEF26 may contribute is a complete mystery and an exciting direction of future study.

Our work has also expanded understanding of ARHGEF26-mediated invasion susceptibility by demonstrating that the protein has serovar and cell line dependent roles during

*Salmonella* infection. Specifically, we demonstrated that while ARHGEF26 influences the susceptibility of HeLa cells to SopB- and SopE-mediated *S.* Typhi invasion, it is critical for SopB- and SopE2-independent *S.* Typhimurium invasion of polarized MDCK cells. This reveals context-dependent roles for ARHGEF26 and raises a number of technical and conceptual questions. At present, we are unsure why this serovar specific interaction occurs, but our data suggest that it is partially dependent on the SopB/SopE/SopE2 paradigm in HeLa cells, as introduction of *S.* Typhimurium SopE2 into *S.* Typhi ablates the effect of *ARHGEF26* knockdown. Further, that serovar specificity occurs in HeLa and MDCK cells, but not in LCLs, raises additional questions about the host factors that contribute to ARHGEF26-mediated invasion.

Bacterial effectors that mimic host proteins often outperform their mammalian counterparts in order to promote pathogenesis [67,68]. That *Salmonella* benefit from both the SopE GEF mimic and ARHGEF26 being present in HeLa cells is a striking exception to this paradigm, but not unheard of. Indeed, previous work has demonstrated that SopE acts cooperatively with the host GEF ARNO (CYTH2) to activate the WAVE complex and induce invasion [23]. As we and others [19,20] have provided evidence that RHOG may be dispensable for invasion, one potential reason for this cooperation may be differences in substrate specificity between SopE and ARHGEF26. Therefore, identifying the GTPase(s) that ARHGEF26 activates during infection represents an important future direction. One potential GTPase is RHOJ (also called TCL), a GTPase that has high homology to CDC42 [69] and is important in *Salmonella* invasion [19]. Alternatively, SopE/ARHGEF26 cooperation could involve a SopE-mediated positive feedback loop that depends on ARHGEF26 GEF activity for full efficiency. We also propose that SopE could have an indirect role in activating ARHGEF26, potentially through modulation of protein-modifying enzymes, as post-translational modifications are important regulators of ARHGEF26 activity [70].

While we were surprised that ARHGEF26 contributes to SopE-mediated HeLa cell invasion, perhaps the most shocking result from pairing host and bacterial genetics was finding that ARHGEF26 is responsible for the entire impact of SopB and SopE2 on *Salmonella* fitness in the enteric fever infection model. The ARHGEF26 requirement fundamentally challenges our understanding of how SopE2 contributes to virulence *in vivo* as it suggests activating small-GTPases to directly induce membrane ruffling is not its primary role. This, paired with our observations that *ARHGEF26* influences SopE-mediated invasion susceptibility in HeLa cells, is fascinating, as it demonstrates that *S.* Typhi and *S.* Typhimurium opt to utilize a host and pathogen protein with near identical functions (ARHGEF26 and SopE/SopE2) to promote invasion and fitness.

In addition to representing fascinating biology, the combination of the above observations also raises concerns about using a single model, particularly HeLa cells, to study *Salmonella* processes. For instance, our murine enteric fever model, in which *S.* Typhimurium had a SPI-1 dependent fitness deficit in *Arhgef26*$^{-/-}$ mice (Fig 7C), are more consistent with our results from H2P2 (Fig 1F), siRNA in LCLs (Fig 2D), and polarized MDCK cells (Fig 6A), than our HeLa cell data (Fig 3D). However, while these cell culture models were better at predicting that ARHGEF26 can impact *S.* Typhimurium fitness than HeLa cells, only the HeLa cell model demonstrated the complete SopB- and SopE- dependence on ARHGEF26-mediated invasion susceptibility (Fig 3C) that mimics our observations with SopB and SopE2 in the mouse ileum (Fig 7D). This reinforces a recent observation that there are striking differences in the invasion mechanisms observed in canonical tissue culture models compared to those observed *in vivo* [54,71]. Together these data serve as a reminder that host-pathogen biology is more complex than any one strain-cell line interaction, and that comparing multiple models can lead to a more complete and accurate understanding of pathogenesis, as we have demonstrated here. That this more nuanced understanding came from utilizing natural human genetic variation to point us towards critical host factors demonstrates the strength of the approach.

Finally, our results studying IL-8 in HeLa cells and inflammation in the murine colitis model were also unexpected, as no report has shown *ARHGEF26* may promote proinflammatory processes, and in fact, some have speculated that it may be involved in anti-inflammatory processes. This is due to its relatively high levels in M2 macrophages [72] as well as its ability to suppress muramyl dipeptide-induced IL-8 production in NOD2 expressing HEK293 cells [73]. While initially this latter finding appears to conflict with our results, we instead believe it merely reinforces a point implied by our data: ARHGEF26 has highly context dependent roles in regulating inflammation. For instance, we identified that *RHOG* knockdown and overexpression phenocopies the impacts of *ARHGEF26* manipulation on uninfected cytokines levels, but not infected levels. Further, we show evidence that ARHGEF26 likely has GEF-dependent and -independent roles in regulating basal cytokine levels. Thus, this pleotropic protein likely plays a plethora of roles in regulating cytokine secretion.

Given our finding that *Arhgef26*[-/-] mice have substantially reduced inflammation, we were surprised to find that bacterial burden was not affected in the large intestine at late time points during the colitis model (Fig 9F). We propose four hypotheses for why we observe this result. The first is that compared to previous work on inflammation-promoted bacterial burden [64], we used a higher inoculum of *S.* Typhimurium and that at this dose *S.* Typhimurium may be able to overcome modest microbiota recolonization. The second is that compared to the avirulent *S.* Typhimurium strain used in that study, *Arhgef26*[-/-] mice may be able to produce enough inflammation to prevent microbiota recolonization. The third hypothesis is that while we observed substantially reduced inflammation at two days post infection by pathology scoring, *Arhgef26*[-/-] mice may have altered dynamics of inflammation, rather than outright ablation of inflammation. Thus, at four days post infection, *Arhgef26* knockout mice may have elevated inflammation that can clear out recolonizing microbes. The fourth hypothesis is that differences in *Arhgef26* expression across the different gastrointestinal compartments could lead to the observation that *S.* Typhimurium burden is reduced in the ileum, but not the cecum or colon, at four days post infection in the colitis model.

We speculate that there are two mechanisms by which ARHGEF26 could contribute to inflammation in the mouse gut. First, ARHGEF26 could impact proinflammatory cytokine release, as we observed in HeLa cells. Second, as ARHGEF26 has been shown to play a role in cell migration [70,74], we speculate that *Arhgef26*[-/-] mice may have reduced immune cell migration to the gut, leading to improved pathophysiology. Determining the role of ARHGEF26 during inflammation could have interesting implications on human health, as inhibiting ARHGEF26 and/or RHOG could be a means of reducing inflammation-driven disease.

## Methods

### Ethics statement

Work involving human lymphoblastoid cell lines has been reviewed by Duke Institutional Review Board and deemed to not constitute Human Subjects Research (Pro00044583, "Functional genetic screens of human variation using lymphoblastoid cell lines"). Mouse studies were carried out with approval by the Duke Institutional Animal Care and Use Committee (A145-18-06, "Analysis of genes affecting microbial virulence in mice") and adhere to the *Guide for the Care and Use of Laboratory Animals* of the National Institutes of Health.

### Mammalian and bacterial cell culture

HapMap LCLs (Coriell Institute) were cultured at 37°C in 5% $CO_2$ in RPMI 1650 media (Invitrogen) supplemented with 10% FBS (Thermo-Fisher), 2 μM glutamine, 100 U/mL penicillin-G, and 100 mg/mL streptomycin. HeLa cells (Duke Cell Culture Facility) and HEK293T (Duke Cell

Culture Facility) were grown in high glucose DMEM media supplemented with 10% FBS, 1mM glutamine, 100 U/mL penicillin-G, and 100mg/mL streptomycin. Cells used for *Salmonella* gentamicin protection assays were grown without antibiotics for at least one hour prior to infection.

All *Salmonella* strains are derived from the *S*. Typhimurium strain 14028s or *S*. Typhi strain Ty2 and are listed in Table B in S1 Tables, and all plasmids are listed in Table C in S1 TablesC. All knockout strains were generated by lambda red recombination [75]. For infection of cells or mice, bacteria were grown overnight in LB broth (Miller formulation, BD), subcultured 1:33 in 1mL cultures, and grown for an additional two hours and forty minutes at 37˚C shaking at 250 RPM. Strains with temperature sensitive plasmids were grown at 30˚C and plasmids removed at 42˚C. Ampicillin was added to LB at 100 µg/mL, kanamycin at 50 µg/mL, chloramphenicol at 20µg/mL.

## DLG1 knockout in MDCK cells

DLG1 was knocked out using LentiCRISPR v2 Blast (Addgene #83480) in MDCK cells. The gRNA used, ATTGGTCAACACAGATAGCT, was designed with CRISPOR for SpCAS9 (NGG) [76]. HEK293FT cells were transfected using a standard Ca+ phosphate protocol to package the lentivirus. DNA was mixed in the following ratios: lentiCRISPR v2 (6 µg), pMD2. G (0.8 µg), and pSPAX2 (4 µg) pMD2.G and pSPAX2 were a gift from Didier Trono, EPFL, Lausanne, Switzerland (Addgene plasmids 12259 and 12260). Cell culture media was changed the following day and lentivirus particles were harvested 48 hours after transfection.

MDCK cells were infected with lentivirus particles overnight. The following day, the infection medium was removed and replaced with complete medium containing 10 µg/ml blasticidin to select for Cas9-expressing cells. Total cell lysates were subjected to WB analysis for protein expression using an anti-DLG1 antibody (Santa Cruz, Clone 2D11 (sc-9961)). Single cell colonies were isolated by serial dilution.

## MDCK cell polarization

MDCK cells were propagated on TC treated plastic with complete DMEM media, and split using 0.25% Trypsin-EDTA (Sigma-Aldrich). To polarize MDCK cells, 125,000 cells were plated per well on untreated glass in a chamber slide (Thermo Fisher #154534) in complete DMEM without antibiotics or phenol red. On days two and four, media was changed. After the media change on day four, cells were used for assays.

## *Salmonella* flow cytometry infection assays

Infection assays were performed as previously described [17]. Briefly, cells were infected with *S*. Typhimurium (LCLs: MOI 30, 60 minutes infection. HeLa: MOI 1, 30 minute infection. MDCK: MOI 30, 30 minute infection, unless otherwise noted) or *S*. Typhi (LCLs: MOI 10, 60 minute infection. HeLa: MOI 30, 30 minute infection, MDCK: MOI 100, 30 minutes, unless otherwise noted). Post infection, cells were treated with 50 µg/mL gentamicin. Two hours post infection, IPTG was added to induce GFP expression. Three hours and fifteen minutes post infection, cells were stained with 7-aminoactinomycin D (Biomol) and analyzed on a Guava Easycyte Plus Flow Cytometer (Millipore). Percent infected host cells was measured by quantifying the percent of GFP+ cells.

## Cellular GWAS screen

Phenotypic screening in H2P2 on 528 LCLs and family-based GWAS analysis was performed using QFAM-parents with adaptive permutation in PLINK v1.9 [77] as previously described

[17]. All analyzed GWAS data is available through the H2P2 web atlas (http://h2p2.oit.duke.edu/H2P2Home/) [17]. QQ plots were plotted using quantile-quantile function in R.

## Dual luciferase assay

The ARHGEF26 locus identified by H2P2 was cloned from the heterozygote HG02860 (population = Gambian and Western Divisions in the Gambia) into the pBV-Firefly Luciferase plasmid [39] by cut and paste cloning. The plasmid map is available here: https://benchling.com/s/seq-2427vsVPRqxsgOj5NM7P. Firefly luciferase plasmids and the Renilla luciferase plasmid pRL-SV40P [39] were co-transfected at a ratio of 50:1 into HeLa cells using the Lipofectamine 3000 kit (Thermo) according to manufacturer instructions. 48 hours post transfection, cells were lysed and analyzed for luciferase activity using the Dual-Luciferase Reporter Assay System (Promega). Luciferase activity was measured by a Synergy H1 plate reader (BioTek).

## siRNA knockdown and knockdown confirmation

LCL (HG01697) knockdown was achieved by plating at 250,000 cells/well in a six well TC treated dish in 500 μL of Accell media (Dharmacon) with either non-targeting Accell siRNA #1 or an Accell *ARHGEF26* SMARTpool (1 μM total siRNA; Dharmacon). After 3 days, cells were resuspended in RPMI at 50,000 cells/well in a 96 well non-TC treated dish.

HeLa knockdown was performed using the following siRNA: siGenome Non-Targeting #5 or a siGENOME SMARTpool targeting *ARHGEF26*, *SCRIB*, *RHOG*, or *DLG1* (Horizon). siRNA were transfected into HeLa cells using the RNAi Max kit (Thermo) according to manufacturer instructions. Assays were performed forty-eight hours post transfection as described above.

Simultaneously, knockdown was confirmed in each experiment by qPCR (S1F Fig). Briefly, RNA was harvested using a RNeasy kit (Qiagen), cDNA was generated with iScript (Bio-Rad), and qPCR was performed by using iTaq Universal Probes Supermix (Bio-Rad) and a QuantStudio 3 thermo cycler (Applied Biosystems). Primers are listed in Table D in S1 Tables. The cycling conditions were as follows: 50˚C for 2 minutes, 95˚C for 10 minutes, and 40 cycles of 95˚C for 15 seconds followed by 60˚C for 1 minute. All qPCR was run in technical duplicate or triplicate. The comparative threshold cycle ($C_T$) was used to quantify transcripts, with the ribosomal 18s gene (RNA18S5) serving as the housekeeping control. $\Delta C_T$ values were calculated by subtracting the $C_T$ value of the control gene from the target gene, and the $\Delta\Delta C_T$ was calculated by subtracting the nontargeting siRNA $\Delta C_T$ from the targeting siRNA $\Delta C_T$ value. Fold change represents $2^{-\Delta\Delta CT}$.

## *ARHGEF26 and RHOG* overexpression plasmids

*ARHGEF26 and RHOG* overexpression plasmids (Table C in S1 Tables) were transformed using the Lipofectamine 3000 kit (Thermo) according to manufacturer instructions. Most plasmids were generated in previous work [44,45], and all remaining plasmids were generated through site-directed mutagenesis (QuickChange Lightning, Agilent) or cut-and-paste cloning. Assays using overexpression plasmids were performed twenty-four hours post transfection.

## Overexpression microscopy

Cells were fixed for thirty minutes in 4% paraformaldehyde + 1% glutaraldehyde and blocked for thirty minutes in a 5% normal donkey serum, 0.2% saponin, PBS solution. Cells were

incubated overnight at 4˚C with an anti-myc antibody (Developmental Studies Hybridoma Bank, 9e10), followed by secondary staining using Alexa Fluor secondary anti-mouse antibody (Thermo). Anti-Myc (9e10) was deposited to the DSHB by Bishop, J.M. (DSHB Hybridoma Product 9e10). Actin staining was performed using Alexa Fluor 647 Phalloidin (Thermo) according to manufacturer instructions. Micrographs were taken using an AMG EVOS microscope.

## Microscopy MDCK invasion assays

MDCK cells were polarized as described above and infected with *S.* Typhimurium (MOI 30) or *S.* Typhi (MOI 100). Thirty minutes post infection, wells were washed once with complete DMEM and media was replaced. Gentamicin (50 μg/mL) and IPTG were both added to kill remaining extracellular bacteria and induce GFP. One and a half hours post infection, 1 drop of the NucBlue Live ReadyProbes Reagent (Thermo) was added to each well to stain the nucleus. Two hours post infection, cells were washed once with PBS, fixed for 30 minutes in 4% paraformaldehyde, and mounted. Microscopy was performed using the 63x objective on a Zeiss LSM 510 confocal microscope. Z-stacks were compressed into Z-projections by max intensity using Fiji [78].

## Phosphoinositide dot blot assays

Twenty-four hours before transfection, 1,500,000 HeLa cells were plated on a 10-cm dish or 1,000,000 HEK293T cells were plated in three separate wells of a six-well dish. Cells were transfected as above, but to normalize for expression, 1μg of AKT-PH-GFP was diluted with 7μg vector. Twenty-four hours later, cells were washed, and directly scraped into lysis buffer (50mM Tris, pH 7.6, 150mM NaCl, 1% Triton X-100, 5mM $MgCl_2$, cOmplete Mini protease inhibitor cocktail (Sigma)), and incubated at 4˚C for 30 minutes. PIP strips (Echelon Biosciences) were blocked using Odyssey Blocking Buffer (Licor) or Intercept Blocking Buffer (Licor). Samples were cleared by centrifugation and diluted 1:25 into blocking buffer before addition to the PIP strips. After one hour incubation with rocking at room temperature, PIP strips were washed 3 times with PBS-T and incubated with an anti-GFP primary antibody (Novus, NB600-308). After one hour rocking at room temperature, PIP strips were washed three times with PBS-T and stained with a IRDye donkey anti-rabbit secondary antibody (Licor). After thirty minutes, strips were washed three times with PBS-T, once with PBS, and imaged on a LI-COR Odyssey Classic.

## Analysis of hela cytokine production

For siRNA experiments, two days post transfection media was changed 2 hours before infection. HeLa cells were then infected with late log phase bacteria (*S.* Typhimurium: MOI 1; *S.* Typhi: MOI 30) for 60 minutes. After 60 minutes, gentamicin was added and bacteria were returned to 37˚C incubator for 5 hours. Six hours post infection supernatants were collected. For overexpression experiments, media was changed 18–24 hours post infection. Six hours after the media change, supernatants were collected. Supernatants were stored at -80˚C until use. Cytokine concentrations were determined using a human IL-8 DuoSet ELISA kit (R&D Systems).

## Mouse infections

C57BL6/J mice were obtained from JAX and housed in barrier cages in the Duke University's Division of Laboratory Animal Resources husbandry facility. Following arrival at the Duke

University's Division of Laboratory Animal Resources husbandry facility, *Arhgef26*[-/-] mice [55] were rederived as specific pathogen free mice by embryo transplantation. Mice were fed rodent diet 5053 chow. Breeding of mice was carried out by the Duke DLAR Breeding Core.

For the enteric fever model of infection, age and sex matched 7–16 week old C57BL/6J or *Arhgef26*[-/-] mice were fasted for 12 hours prior to infection and treated with a 100μL of a 10% sodium bicarbonate solution 30 minutes prior to infection. Bacteria were grown as described above, washed, and resuspended in PBS at a concentration of $1x10^9$ bacteria/mL, and 100μL were administered to the mice for an estimated final dose of $1x10^8$ bacteria/mouse. Inoculum was confirmed by plating for CFUs. All mice were monitored daily for changes in morbidity. Mice were euthanized by $CO_2$ asphyxiation four days post infection and tissues were harvested, weighed, homogenized, and plated for CFU quantification.

For the colitis model of infection [10], age and sex matched 7–16 week old C57BL/6J or *Arhgef26*[-/-] mice were fasted four hours before treatment with 20μg of streptomycin (Sigma) in 75μL of sterile water 24 hours before infection. Food was returned until four hours before infection, when they were fasted again. Thirty minutes before infection, mice received 100μL of a 10% sodium bicarbonate solution. Bacteria, grown as described above, were washed and resuspended in PBS at a concentration of $1x10^{10}$ bacteria/mL, and 100μL were administered to the mice for an estimated final dose of $1x10^9$ bacteria/mouse. Food was returned four hours after infection. Inoculum was confirmed by plating for CFUs. All mice were monitored daily for changes in morbidity. For inflammatory phenotypes, two days post infection, mice were euthanized by $CO_2$ asphyxiation, and tissues were removed and either weighed and plated for CFUs as described above or prepared for histopathologic examination. For late stage CFU quantification, mice were harvested identically four days post infection.

Cecal and colon tissues were fixed 48–72 hours in 10% neutral buffered formalin, processed routinely, embedded in paraffin, cut at 5mm and stained with hematoxylin and eosin. Tissues were evaluated in a masked fashion by a board-certified veterinary pathologist (JIE) with allocation group concealment. Tissues were scored using a semi-quantitative grading system of multiple parameters and anatomic compartments (lumen, surface epithelium, mucosa, and submucosa) to assign summary pathologic injury scores [79].

The histopathologic scoring was (scores in parenthesis). (a) Lumen: empty (0), necrotic epithelial cells (scant, 1; moderate, 2; dense, 3), and polymorphonuclear leukocytes (PMNs) (scant, 2; moderate, 3; dense, 4). (b) Surface epithelium: No pathological changes (0); mild, moderate, or severe regenerative changes (1, 2, or 3, respectively); patchy or diffuse desquamation (1 or 2); PMNs in epithelium (1); and ulceration (1). (c) Mucosa: No pathological changes (0); rare (<15%), moderate (15 to 50%), or abundant (>50%) crypt abscesses (1, 2, or 3, respectively); presence of mucinous plugs (1); presence of granulation tissue (1). (d) Submucosa: No pathological changes (0); mononuclear cell infiltrate (1 small aggregate, <1 aggregate, or large aggregates plus increased single cells) (0, 1, or 2, respectively); PMN infiltrate (no extravascular PMNs, single extravascular PMNs, or PMN aggregates) (0, 1, or 2, respectively); mild, moderate, or severe edema (0, 1, or 2, respectively).

## Statistics

All statistics were performed using Graphpad Prism 9 or Microsoft Excel, unless otherwise noted. Bars and central tendency representations as well as p-value calculations are described in all figure legends. Where noted, inter-experimental noise was removed prior to data visualization or statistical analysis by standardizing data to the grand mean by multiplying values within an experiment by a constant (average of all experiments divided by average of specific experiment). Data points were only excluded if technical failure could be proven (*i.e.* failed

RNAi knockdown measured by qPCR), or if identified by an outlier test where indicated. If datapoints were removed by an outlier test, the original results are reported in the figure legend.

## Supporting information

**S1 Table.** A: Genes used for stratification. B: Bacterial strains used in this study. C: Plasmids used in this study. D: Oligonucleotides used in this study.
(DOCX)

**S1 Fig. Natural variation in ARHGEF26 and ARHGEF26 knockdown associate with reduced Salmonella uptake.** (A) Stratified QQ plots of host cell infection reveal that, in addition to rs993387, multiple SNPs in *ARHGEF26* (blue diamonds) associate with *S*. Typhi or *S*. Typhimurium infection in H2P2 at lower p-values than expected by chance. (B) Removing SNPs in *ARHGEF26* from the QQ plot removes any deviation of SNPs in SPI-1 associated genes from p-values expected by chance. (C) rs993387 association was observed across all four populations studied in H2P2 (IBS, Iberians from Spain; GWD, Gambian from the Western Divisions of The Gambia; ESN, Esan in Nigeria; KHV, Kinh in Ho Chi Minh City, Vietnam). Each dot represents a single LCL line averaged across three independent experiments. Black bar represents the median. LCLs in each population follow the trend of TT < GT < GG, except in GWD and ESN for Typhi where only 1 or 2 GG individuals were assayed. (D) A luciferase reporter system was generated to assess whether the rs993387 locus has enhancer activity. A roughly 5kb region was cloned from a heterozygous individual (HG02860, GWD) into pBV-Luc upstream of a minimal promoter and the firefly luciferase gene. Performing a dual luciferase experiment in HeLa cells revealed enhanced luciferase expression with the rs993387 locus. Bars represent the relative firefly luciferase/renilla luciferase activity, with vector set to 1. P-Values generated from one-way ANOVA with Tukey's multiple comparisons test on the log transformed values. (E) siRNA targeting *ARHGEF26* results in reduced expression in LCLs (HG01697, IBS). (F) siRNA targeting *ARHGEF26*, *DLG1*, *SCRIB*, and *RHOG* results in reduced expression in HeLa cells. Lines in E and F represent median fold change. Fold change is calculated as $2^{-\Delta\Delta CT}$ using RNA18S5 as a housekeeping control gene.
(DOCX)

**S2 Fig. ARHGEF26 does not demonstrate phosphoinositide binding.** ARHGEF26, RHOG, and the GFP-AKT-PH constructs were overexpressed in either HeLa or HEK293T cells. Cells were lysed, and protein extract was diluted in the listed blocking buffer before incubation on PIP strips. No robust ARHGEF26 signal on dotted phosphoinositide species could be detected following immunostaining. Key in bottom right displays location of each phosphoinositide species using a PIP strip incubated with GFP-AKT-PH (also shown in top left) as an example.
(DOCX)

**S3 Fig. Confirmation of the DLG1 knockout MDCK cells.** DLG1 knockout was confirmed by both immunofluorescence (DLG1 in Gray/Red, Nucleus in Blue, Left) and immunoblotting whole cell lysates (Right). Anti-DLG1 antibody (Santa Cruz Clone 2D11 (sc-9961)) used to detect DLG1.
(DOCX)

**S1 Data. Breakdown of pathology scores for C57BL/6J and Arhgef26$^{-/-}$ mice.**
(XLSX)

## Acknowledgments

We would like to thank Alyson Barnes, Ben Schott, Alejandro Antonia, Sarah Jaslow, Kelly Pittman, Rachel Keener, and all other past and present members of the Ko Lab for their support throughout this project. In particular, we thank Dr. Kyle Gibbs for his thorough editing of this manuscript and frequent contributions to experimental design, and Caroline Anderson for experimental assistance. We thank Dr. Keith Burridge for early discussion on ARHGEF26 and a gift of *Arhgef26*$^{-/-}$ mice. We thank Kristin Cleveland and Duke DLAR Breeding Core personnel for breeding and maintenance of mouse lines. We thank Wolf-Dietrich Hardt and Barthel Manja for providing the SopB and SopE2 bacterial expression plasmids. The *S.* Typhimurium *sopE*::*tet* strain was a gift from Dr. Heather Felise. We also thank Dr. Stacy Horner and the Duke MGM Department for use of equipment. The Duke University Light Microscopy Core Facility (LMCF) and Dr. Lisa Cameron were instrumental in performing confocal microscopy. All schematic images were generated using Biorender.com.

## Author Contributions

**Conceptualization:** Jeffrey S. Bourgeois, Rafael Garcia-Mata, Dennis C. Ko.

**Formal analysis:** Jeffrey S. Bourgeois, Liuyang Wang, Jeffrey Everitt.

**Funding acquisition:** Dennis C. Ko.

**Investigation:** Jeffrey S. Bourgeois, Agustin F. Rabino, Jeffrey Everitt, Monica I. Alvarez.

**Methodology:** Jeffrey Everitt.

**Resources:** Agustin F. Rabino, Sahezeel Awadia, Erika S. Wittchen, Rafael Garcia-Mata.

**Supervision:** Dennis C. Ko.

**Visualization:** Agustin F. Rabino.

**Writing – original draft:** Jeffrey S. Bourgeois.

**Writing – review & editing:** Jeffrey S. Bourgeois, Rafael Garcia-Mata, Dennis C. Ko.

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
