## [Decision Letter · Decision Letter 0]

1 Feb 2021

Dear Dennis,

Thank you very much for submitting your manuscript "ARHGEF26 enhances Salmonella invasion and inflammation in cells and mice" for consideration at PLOS Pathogens. As with all papers reviewed by the journal, your manuscript was reviewed by members of the editorial board and by several independent reviewers. In light of the reviews (below this email), we would like to invite the resubmission of a significantly-revised version that takes into account the reviewers' comments.

There are quite a few experiments suggested by the three reviewer's to clarify some of your findings. Each point they make is valid. Please pay particular attention to their comments related to the in vivo studies and the recruitment of ARHGEF26 during bacterial-mediated invasion. After reading the manuscript, I have a couple of additional comments.

(1) Your method of testing "invasion" is unusual. It strictly does not measure invasion (total number of bacteria that were internalised in the monolayer) but rather the percentage of cells that are infected. And it is done at 3 h 15 min post-infection, which is well after the initial invasion event. By this time, Salmonella would have undergone 1-2 rounds of replication inside of cells. As pointed out by one reviewers, your differences in "invasion" are small (<15%). Please confirm whether you get similar results with an assay that quantifies the total number of bacteria that have been internalized e.g. gentamicin protection assay.

(2) The streptomycin-pretreatment mouse model is not a model of gastroenteritis. Rather it should be termed a mouse colitis model. See https://pubmed.ncbi.nlm.nih.gov/21343352/ for clarification. Please correct this throughout, including in figures.

We cannot make any decision about publication until we have seen the revised manuscript and your response to the reviewers' comments. Your revised manuscript is also likely to be sent to reviewers for further evaluation.

Sincerely,

Leigh Knodler

Guest Editor

PLOS Pathogens

Nina Salama

Section Editor

PLOS Pathogens

Kasturi Haldar

Editor-in-Chief

PLOS Pathogens

orcid.org/0000-0001-5065-158X

Michael Malim

Editor-in-Chief

PLOS Pathogens

orcid.org/0000-0002-7699-2064

Dear Dennis,

There are quite a few experiments suggested by the three reviewer's to clarify some of your findings. Each point they make is valid. Please pay particular attention to their comments related to the in vivo studies and the recruitment of ARHGEF26 during bacterial-mediated invasion. After reading the manuscript, I have a couple of additional comments.

(1) Your method of testing "invasion" is unusual. It strictly does not measure invasion (total number of bacteria that were internalised in the monolayer) but rather the percentage of cells that are infected. And it is done at 3 h 15 min post-infection, which is well after the initial invasion event. By this time, Salmonella would have undergone 1-2 rounds of replication inside of cells. As pointed out by one reviewers, your differences in "invasion" are small (<15%). Please confirm whether you get similar results with an assay that quantifies the total number of bacteria that have been internalized e.g. gentamicin protection assay.

(2) The streptomycin-pretreatment mouse model is not a model of gastroenteritis. Rather it should be termed a mouse colitis model. See https://pubmed.ncbi.nlm.nih.gov/21343352/ for clarification. Please correct this throughout, including in figures.

With my best wishes, Leigh

Reviewer's Responses to Questions

**Part I - Summary**

Reviewer #1: It has been known for many years that pathogenic Salmonella serovars manipulate host cell GTPases to drive bacterial internalization into non-phagocytic epithelial cells. Previous work has suggested that one of these GTPases is RhoG, activated by the host GEF ARHGEF26 (also known as SGEF) in response to phosphoinositide changes induced by the bacterial effector protein SopB. Here the authors identified ARHGEF26 in a focused genetic screen for natural genetic variants that enhance the infection of lymphoblastoid cells by S. Typhi or S. Typhimurium. Using a variety of in vitro and in vivo (mouse) assays, they demonstrate a small but statistically significant impact of ARHGEF26 on both bacterial internalization and the induction of inflammatory signaling (IL-8 production). In vitro, knockdown of ARHGEF26 appears to selectively affect S. Typhi as opposed to S. Typhimurium, presumably due to differences in the arrays of secreted effector proteins produced by each serovar. In contrast, the most physiologically significant difference is seen in mouse models of S. Typhimurium infection, where knockout of ARHGEF26 inhibits colonization of the ileum (in an enteric fever model) or inflammation, but not colonization in the colon (in a gastroenteritis model).

While the data certainly suggest a potential genetic basis for differential susceptibility to Salmonella infection, no clear unifying mechanism is identified. There are many “plausible hypotheses” but no firm conclusions. The in vitro data indicate that RhoG is not involved in invasion (contradicting an earlier study from the Galan lab), but that the bacterially encoded Rho-GEF SopE is, however no link between ARHGEF26 and SopE or its downstream targets Rac1 and Cdc42 is established. The data also implicate two scaffolding proteins, DLG1 and SCRIB, which are involved in the establishment and maintenance of epithelial junctions, however how they contribute to invasion is never defined, or even explored.

Most of the phenotypic effects of ARHGEF26 depletion in vitro are relatively small, suggesting that, whatever its function, it is not a primary driver of invasion or inflammation but may contribute in some secondary way.

The effects observed in vivo are somewhat larger, but confusing and somewhat contradictory. While depletion of ARHGEF26 had no effect at all on S. Typhimurium invasion in cultured cells, it had a significant effect in the ileum during oral infection of ARHGEF26-deficient mice with S. Typhimurium. It is possible that the interacting partners DLG1 and SCRIB have more important roles in the context of a polarized epithelium (i.e. the ileum) than in non-polarized cells in culture, but this issue is never explored.

In a different model of oral infection (streptomycin pre-treatment) where colonization occurs primarily in the colon, not the ileum, the data indicate no difference in colonization but a significant difference in inflammation (reduced in the absence of ARHGEF26). Although it is possible that regional differences in the intestinal epithelium itself (or regional expression of ARHGEF26) could account for this, the use of two different models of infection makes the result impossible to interpret from a mechanistic perspective.

Reviewer #2: Through the GWAS screen, Bourgeois et al. identified ARHGEF26 (also known as SGEF) as a host factor that up-regulates Salmonella invasion. In vitro invasion assays using HeLa cells, they found that ARHGEF26 contributes to SopB- and SopE-mediated S. Typhi (but not S.Typhimurium) invasion. They examined the molecular mechanism how ARHGEF26 activity is regulated, focusing on the ARHGEF interactors, DLG1 and SCRIB, as well as by the ARHGEF26 domain analysis. They further showed the possible role of ARHGEF26 in Salmonella-induced inflammation by monitoring IL-18 secretion. Finally, they demonstrated the contribution of ARHGEF26 to S. Typhimurium fitness using the mouse model.

Overall, the manuscript is well-written and gives new insights into a role of ARHGEF26 in SopE-mediated Salmonella invasion and inflammation, although there are some inconsistencies between results depending on the experimental systems (ex. HeLa cells vs LCLs) as the authors discussed. For the disagreement with a previous report, I think that some results need to be more clarified by conducting additional experiments.

Reviewer #3: In this manuscript the author have investigated the role of ARHGEF26 in response to Salmonella Typhi and Salmonella Typhimurium infection. They demonstrate that the effector proteins SopE and SopB of S. Typhi are required for efficient ARHGEF26-dependent invasion in HeLa cells, contrary to S. Typhimurium effectors SopE2 and SopB which appeared not to require ARHGEF26 to invade HeLa cells. They further demonstrate that activation of ARHGEF26 can induce inflammatory responses, and that ARHGEF26 plays a critical role in S. Typhimurium pathogenesis in both the enteric fever murine model and the murine gastroenteritis infection model.

Strengths:

Natural genetic variations in ARHGEF26 were identified in a cellular GWAS screen that contribute to susceptibility to Salmonella Typhi and Typhimurium invasion.

Demonstrated new roles of ARHGEF26 during Salmonella invasion.

Strong in vitro and in vivo data with a great deal of experimental repeats.

Weaknesses:

Some controls that will strengthen the conclusions are missing.

Additional time points in the in vivo experiments should be included.

**Part II – Major Issues: Key Experiments Required for Acceptance**

Reviewer #1: 1. Fig. 2 shows a very small (<15%) reduction in invasion of LCLs by both S. Typhi and S. Typhimurium after knockdown of ARHGEF26. While Fig. 3 shows a slightly higher inhibition of invasion in Hela cells by S. Typhi (~30%), there is no effect on S. Typhimurium. How do the authors explain this discrepancy? Do both cell types express ARHGEF26 at similar levels? What about DLG1 or SCRIB? Are there differences in expression?

2. Even in Hela cells, knockdown of ARHGEF26, DLG1 or SCRIB yielded only small differences in invasion (Fig. 4). Given the important roles of DLG1 and SCRIB in epithelial junction formation, it is surprising that the authors did not examine invasion in a polarized epithelial cell model (e.g. Caco-2 or HT-29 cells).

3. Dissection of ARHGEF26 suggests that interaction with DLG1 and/or SCRIB is important to its function. Do these interactions stimulate the GEF activity of ARHGEF26?

4. As mentioned above, the mouse models yield confusing and contradictory data. One possible reason for this is that the ARHGEF26 knockout is not tissue-specific, and the data could reflect altered responses in the epithelium, stromal cells or infiltrating lymphocytes. It could also reflect differential expression of ARHGEF26 or splice variants in the ileum vs. cecum or colon. Neither of these issues is addressed, and the results are therefore impossible to interpret.

Reviewer #2: (1) The in vitro invasion assay using HeLa cells demonstrated that ARHGEF26 contributes to both SopB- and SopE-mediated S. Typhi invasion (Figure 3), while the previous work (Ref 7) showed that ARHGEF26 is involved in SopB-, but not SopE-mediated (S. Typhimurium) invasion. As the genetic background of Salmonella strains (as well as cell-lines) are different between the two works, it is important to ensure that ARHGEF26 works solely on the SopB/SopE axis in the strains used in this study. The double knockout (ΔsopB ΔsopE) S. Typhi strain (which should give no difference between NT and siARHGEF26) needs to be included in Figure 3D as well as in Figure 4C). This strain may give the passive invasion as observed with ΔprgH (Figure 7C), but should be a required control.

(2) According to Awadia et al. (Ref 48), detail of the molecular interaction between ARHGEF26 and DLG1/SCRIB has been analyzed. DLG1- and SCRIB-binding regions seem to locate just upstream of the DH domain of ARHGEF26. I wonder if the 415-871 construct in Figure 5A possesses the ability to interact with DLG1- and SCRIB. As the authors propose that SCRIB-DLG1-ARHGEF26 complex guides the GEF to the plasma membrane (lines 699-700), the interaction should be disrupted experimentally by introducing mutations in the binding regions to see if the interaction is essential for enhancing invasion.

(3) For clarifying the role of ARHGEH26 in Salmonella invasion, it is crucial to demonstrate the recruitment of the protein to the site of invasion. Infection experiments can be conducted using HeLa cells expressing Myc-tagged ARHGEF26 and its derivatives, not only showing transfection-induced ARHGEF26-positive membrane ruffles (Figure 5DE).

Reviewer #3: 1. The authors show that ARHGEF26 is not required for S. Tm SopB-dependent invasion, and speculate that a modest difference in sequence between SopB of S. Tm and S. Typhi might account for the differential ARHGEF26 requirement. Have the authors considered complementing a S. Tm DsopE2/sopB double mutant with S. Typhi SopB and/or SopE to investigate whether the ARHGEF26 requirement is dependent on SopB and/or SopE of S. Typhi and not caused by other differences in the invasion mechanisms of S. Typhi and S. Tm? The reverse can also be achieved; complement S. Typhi mutants with S. Tm SopB and/or SopE2 to clearly pinpoint whether these effector proteins contribute (or not) to ARHGEF26-dependent invasion in HeLa cells.

2. The in vivo role of ARHGEF26 was determined by infecting Arhgef26-/- mice using two different mouse models, the enteric fever murine model and the murine gastroenteritis infection model.

Arhgef26-/-mice have reduced bacterial numbers in ileum and spleen (Fig 7), and this is dependent on a functional T3SS-1, since a DprgH mutant no longer showed reduced bacterial numbers in Arhgef26-/-mice compared to wild type mice. A DsopB mutant, however, still had significantly lower numbers in the ileum, suggesting S. Tm SopB does not contribute to ARHGEF26-dependent bacterial colonization. It is unclear why the authors state that this is dependent on SopB (lines 601). Is this conclusion solely based on the reduced ratio (0.04) compared to mice infected with wild type Salmonella (0.004)? What is the bacterial load of the DsopB mutant in the spleen of wild type and Arhgef26-/- mice? And does SopE play a significant role in the murine enteric fever model?

3. In the gastroenteritis model is there a reduction in cytokine (eg. KC) production that would correlate with the findings in HeLa cells?

It has been demonstrated that Salmonella-induced inflammation is beneficial for Salmonella to grow to high numbers. Does the reduction in total pathology scores result in reduced bacterial colonization in the Arhgef26-/- mice at later days post infection (days 3 and 4)? And do the effector proteins SopB and/or SopE contribute this ARHGEF26-dependent inflammation?

**Part III – Minor Issues: Editorial and Data Presentation Modifications**

Reviewer #1: (No Response)

Reviewer #2: (1) I wonder why the ΔsopB S. Typhi strain shows the higher invasion late than the wild-type strain in HeLa infection experiments (Figure 3D, Figure 4CD). Are there any reasonable explanations?

(2) The authors did not get experimental data supporting that RhoG has any roles in Salmonella invasion and in “infection-induced” IL-8 production. I wonder if the authors think that the genetic diversity of SopE in the Salmonella strains can cause the discrepancy between studies (lines 720-722).

(3) I assume that Figure 4 experiments were conducted using HeLa cells, but I cannot find the description.

(4) Description of Figure S1 (D) is missing in the legend.

Reviewer #3: 1. In Figure 3D, it seems that the DsopB strain is more invasive compared to the S. Typhi wild type strain Ty2 in both NT and siARHGEF26 HeLa cells. Is this difference significant, and if so, how can this be explained? It shows that SopE-dependent invasion is more efficient (when SopB is deleted), also in the absence of ARHGEF26.

2. Would a DsopB/sopE S. Typhi mutant invade NT and siARHGEF26 cells similarly? Or are there other effectors also required for ARHGEF26-dependent invasion?

3. Figure 6A shows that silencing of ARHGEF26 reduces IL-8 production in unstimulated cells by approximately 2-fold. SiARHGEF26 cells infected with S. Typhi of S. Tm also had reduced IL-8 production. Is this moderately reduced IL-8 production a result of the lower basal IL-8 production, or is it an effect of Salmonella-induced inflammation. Did the authors correct for the IL-8 reduction in unstimulated cells?

PLOS authors have the option to publish the peer review history of their article (what does this mean?). If published, this will include your full peer review and any attached files.

Reviewer #1: No

Reviewer #2: No

Reviewer #3: No
---

## [Editor Report · Decision Letter 1]

14 Jun 2021

Dear Dennis,

We are pleased to inform you that your manuscript 'ARHGEF26 enhances Salmonella invasion and inflammation in cells and mice' has been provisionally accepted for publication in PLOS Pathogens.

Best regards,

Leigh Knodler

Guest Editor

PLOS Pathogens

Nina Salama

Section Editor

PLOS Pathogens

Kasturi Haldar

Editor-in-Chief

PLOS Pathogens

orcid.org/0000-0001-5065-158X

Michael Malim

Editor-in-Chief

PLOS Pathogens

orcid.org/0000-0002-7699-2064
---

## [Editor Report · Acceptance letter]

1 Jul 2021

Dear Ko,

We are delighted to inform you that your manuscript, "ARHGEF26 enhances Salmonella invasion and inflammation in cells and mice," has been formally accepted for publication in PLOS Pathogens.

Best regards,

Kasturi Haldar

Editor-in-Chief

PLOS Pathogens

orcid.org/0000-0001-5065-158X

Michael Malim

Editor-in-Chief

PLOS Pathogens

orcid.org/0000-0002-7699-2064